# Robust learning of halfspaces under log-concave marginals

**Jane Lange**[*]
MIT
jlange@mit.edu

**Arsen Vasilyan**[*]
UT Austin
ArsenVasilyan@gmail.com

## Abstract

We say that a classifier is *adversarially robust* to perturbations of norm $r$ if, with high probability over a point $x$ drawn from the input distribution, there is no point within distance $\leq r$ from $x$ that is classified differently. The *boundary volume* is the probability that a point falls within distance $r$ of a point with a different label. This work studies the task of computationally efficient learning of hypotheses with small boundary volume, where the input is distributed as a subgaussian isotropic log-concave distribution over $\mathbb{R}^d$.

Linear threshold functions are adversarially robust; they have boundary volume proportional to $r$. Such concept classes are efficiently learnable by polynomial regression, which produces a polynomial threshold function (PTF), but PTFs in general may have boundary volume $\Omega(1)$, even for $r \ll 1$.

We give an algorithm that agnostically learns linear threshold functions and returns a classifier with boundary volume $O(r+\varepsilon)$ at radius of perturbation $r$. The time and sample complexity of $d^{\tilde{O}(1/\varepsilon^2)}$ matches the complexity of polynomial regression.

Our algorithm augments the classic approach of polynomial regression with three additional steps:

a) performing the $\ell_1$-error regression under noise sensitivity constraints,

b) a structured partitioning and rounding step that returns a Boolean classifier with error $\mathsf{opt} + O(\varepsilon)$ and noise sensitivity $O(r + \varepsilon)$ simultaneously, and

c) a local corrector that "smooths" a function with low noise sensitivity into a function that is adversarially robust.

## 1  Introduction

A predictor is robust to adversarial examples if for most possible inputs, a small perturbation will not cause the input to be misclassified. We define the *boundary volume* as the probability, over the input distribution, that a point is close to the boundary:

$$\mathrm{Boundary}_{\mathcal{D},r}(f) = \Pr_{x \sim \mathcal{D}}[\exists z : \|z\|_2 \leq r \text{ and } f(x) \neq f(x + z)].$$

The sum of the boundary volume and the classification error is a natural upper bound on the standard notion of robust risk:

$$\mathrm{RobRisk}_{\mathcal{D},r}(f) = \Pr_{(x,y) \sim \mathcal{D}}[\exists z : \|z\|_2 \leq r \text{ and } y \neq f(x + z)].$$

In this work, we discuss computationally efficient learning of classifiers with optimal classification error and boundary volume, thus minimizing the robust risk.

---

[*]Alphabetical order.

39th Conference on Neural Information Processing Systems (NeurIPS 2025).

There is a large body of research on adversarial robustness in machine learning, the focus of which is to assess the robustness of classifiers commonly deployed in practice — see [BCM+13, GSS14, SZS+14], as well as [KM18] for an overview of the topic. It is well known that deep networks trained on classic benchmark data sets, such as ImageNet, can be "tricked" into misclassifying a test input by making a perturbation so small that it cannot be detected by humans, and that training robust models is a challenge in practice.

In this work we consider linear threshold functions (halfspaces), one of the most basic and well-known concept classes. As a robustness benchmark, we consider the robustness of a *proper* learning algorithm — one that outputs a hypothesis that is a halfspace. For a halfspace, when the input is distributed according the $d$-dimensional standard Gaussian — or more generally, a subgaussian isotropic log-concave distribution — the probability that an input falls within Euclidean distance $r$ of the classification boundary is $O(r)$; thus a $1 - O(r)$ fraction of inputs are robust to adversarial perturbations of norm $r$. A proper agnostic learner would output a hypothesis with the following guarantees:

- **Agnostic approximation:** $\Pr_{(x,y)\sim\mathcal{D}}[y \neq h(x)] \leq \mathsf{opt} + \varepsilon$, where $\mathsf{opt}$ is the classification error of the best halfspace, and

- **Adversarial robustness:** $\mathrm{Boundary}_{\mathcal{D},r}(h) \leq O(r)$.

However, the state of the art for agnostically learning halfspaces is an improper algorithm — an algorithm that does not output a halfspace, but instead outputs a polynomial threshold function (PTF). Polynomial regression with randomized rounding learns halfspaces over $\mathcal{N}(0, I_d)$ with time and sample complexity $d^{\tilde{O}(1/\varepsilon^2)}$ and satisfies the accuracy guarantee [DGJ+09, KKMS08]. In fact, the efficiency of polynomial regression is due to the inherent robustness of halfspaces — their small surface area! However, there exist degree-$1/\varepsilon^2$ PTFs with boundary volume $\Omega(1)$ even at a small radius of perturbation,[2] and this algorithm makes no guarantee that its output is robust.

No proper agnostic learning algorithm with $2^{o(d)}$ running time for general subgaussian isotropic log-concave distributions is known, and for the Gaussian distribution the state of the art for proper learning is $d^{O(1/\varepsilon^4)}$ [DKK+21]. The complexity of proper learning in these settings is still an open question.We circumvent the possible difficulty of proper learning by giving an improper algorithm that still satisfies both the accuracy and robustness properties. Our algorithm has time and sample complexity $d^{\tilde{O}(1/\varepsilon^2)}$, matching the run-time of the best (improper and non-robust) agnostic learning algorithm [DGJ+09]. Up to polylogarithmic factors in the exponent, this also matches the statistical query lower bound for agnostically learning halfspaces [DKPZ21].

## 1.1 Main result

Our main result is an improper agnostic learner (Algorithm 1) for halfspaces that outputs an adversarially robust hypothesis.

**Theorem 1.1** (Adversarially robust learning of halfspaces, informal version of Theorem 3.1)**.** *Let $\mathcal{D}$ be a distribution over $\mathbb{R}^d \times \{-1, 1\}$ such that the $\mathbb{R}^d$-marginal is subgaussian, isotropic, and log-concave. There is an algorithm that takes a robustness radius $r$, an accuracy parameter $\varepsilon$, and a sample of size $d^{\tilde{O}(1/\varepsilon^2)}$ from $\mathcal{D}$. With high probability, it outputs a hypothesis $h : \mathbb{R}^d \to \{-1, 1\}$ with the following guarantees:*

- *Agnostic approximation:* $\Pr_{(x,y)\sim\mathcal{D}}[h(x) \neq y] \leq \mathsf{opt} + O(\varepsilon)$*, where $\mathsf{opt}$ is the classification error of the best halfspace.*

- *Adversarial robustness:* $\mathrm{Boundary}_{\mathcal{D},r}(h) \leq O(r)$.

## 1.2 Technical overview and intermediate results

Our algorithm and its analysis have three main components, which may be of independent interest. Two of the components solve a relaxation of the robust learning problem, and the third transforms the almost-robust hypothesis into a robust one. The relaxed learner produces a hypothesis with small

---

[2]Consider the PTF $\mathrm{sign}(\prod_{i=1}^{1/\varepsilon^2} x_i)$, which has boundary volume $\Omega(1)$ over $\mathcal{N}(0, I_d)$ at any radius $\geq \varepsilon^2$.

*noise sensitivity* (a notion of boundary volume with random perturbations instead of adversarial perturbations) and small *isolation probability* (the probability that the local noise sensitivity around a point is very high). To formally define these two quantities, we need the following notion:

**Definition** (Local noise sensitivity)**.** *We define the local noise sensitivity $\phi_{f,\eta}(x)$ at noise scale $\eta$ as*

$$\phi_{f,\eta}(x) := \mathop{\mathbb{E}}_{z\sim\mathcal{N}(0,I_d)}[\tfrac{1}{2}|f(x)-f(x+\eta z)|].$$

*We remark that when $f$ is Boolean-valued, this is equivalently*

$$\phi_{f,\eta}(x) := \mathop{\Pr}_{z\sim\mathcal{N}(0,I_d)}[f(x)\neq f(x+\eta z)].$$

We now define the noise sensitivity and the isolation probability:

**Definition** (Distributional noise sensitivity)**.** *The noise sensitivity of a function $f$ at noise scale $\eta$ on distribution $\mathcal{D}$ is defined as*

$$\mathrm{NS}_{\mathcal{D},\eta}(f) = \mathop{\mathbb{E}}_{x\sim\mathcal{D}}[\phi_{f,\eta}(x)]. \tag{1}$$

We remark that when $\mathcal{D} = \mathcal{N}(0, I_d)$, our definition of distributional noise sensitivity differs from the standard definition of Gaussian noise sensitivity.

**Definition** (Isolation probability)**.** *We call a point* isolated *if its local noise sensitivity is over a threshold. The* isolation probability *of a function $f$ relative to a threshold $t$ is*

$$\mathrm{iso}_{\mathcal{D},\eta}(f,t) := \mathop{\Pr}_{x\sim\mathcal{D}}[\phi_{f,\eta}(x)>t].$$

Intuitively, a function can have low distributional noise sensitivity and still not be adversarially robust, if the classification boundary lies near many isolated points scattered throughout the domain. For this reason, after learning a hypothesis with low noise sensitivity, we apply the local correction algorithm which we describe below, which eliminates regions of high local noise sensitivity (isolated points).

### 1.2.1   Local correction of adversarial robustness (Algorithm 3)

The part of our algorithm that transforms an almost-robust hypothesis into a robust one is a *local corrector* for adversarial robustness. Local correctors are part of the family of *local computation algorithms,* or LCAs [RTVX11, ARVX12]: fast randomized algorithms that compute parts of a large object without constructing the object in its entirety. Local correctors, also known as local property reconstructors [ACCL08], are LCAs that evaluate queries to a nearby object that satisfies a desired property; in our case that object is an adversarially robust hypothesis. Our algorithm uses local correction in the fashion of [LRV22]: to "append" a local corrector with a fixed random seed to a non-robust hypothesis in order to *globally* correct the hypothesis.

We analyze a very simple LCA that makes queries to a function and outputs queries to a nearby function with reduced boundary volume. This algorithm estimates the local noise sensitivity at $x$, and changes the label of $x$ if the local noise sensitivity is too high. This is, in essence, the smoothing procedure discussed in [LAG$^+$19, LCWC, CRK19]. We give guarantees on the boundary volume and the error introduced by the smoothing procedure in terms of the relaxed robustness properties of the input function (the noise sensitivity and isolation probability).

**Lemma 1.2** (Local corrector for adversarial robustness, informal version of Theorem C.1)**.** *Let $\varepsilon, \alpha, \beta, r \in (0,1)$. Let $f : \mathbb{R}^d \to \{-1, 1\}$ be a degree-$k$ polynomial threshold function with noise sensitivity $\leq \alpha$ and isolation probability $\leq \beta$. There is an efficient randomized algorithm* ROBUSTNESSLCA *that makes black-box queries to $f$ and answers black-box queries to a function $g : \mathbb{R}^d \to \{-1, 1\}$ that satisfies the following with high probability:*

- *Adversarial robustness:* $\mathrm{Boundary}_{\mathcal{D},r} \leq O(\alpha + \varepsilon)$.

- *Small distance:* $\Pr_{x\sim\mathcal{D}}[g(x) \neq f(x)] \leq O(\beta + \varepsilon)$.

We use this algorithm in a "deterministic" fashion, where all calls to the local corrector share the same random seed, which is good with high probability. When the random seed is good, the guarantees

hold for all $x \in \mathbb{R}^d$.[3] This allows us to append the local corrector and its fixed seed to some degree-$k$ PTF, creating a deterministic robust hypothesis that can be evaluated in $d^{O(k)}$ time.

From the guarantees of the local corrector and the desired guarantees of the overall algorithm, we determine that we must feed the local corrector a PTF of error $\mathsf{opt} + O(\varepsilon)$, noise sensitivity $O(r)$, and isolation probability $O(\varepsilon)$.

### 1.2.2 Polynomial regression under noise sensitivity constraints (Algorithm 2)

Our first step in finding such a PTF is to learn a polynomial with low error, noise sensitivity, and isolation probability in the $\ell_1$-distance regime.

**Theorem 1.3** (Learning a polynomial, informal version of Theorem 3.3)**.** *There is an algorithm with time and sample complexity $d^{\tilde{O}(1/\varepsilon^2)}$ that takes samples from the distribution $\mathcal{D}$ and, with high probability, returns a degree-$\tilde{O}(1/\varepsilon^2)$ polynomial $p$ with the following properties:*

- ***Accuracy:*** $\mathrm{err}(p) \leq \mathsf{opt} + O(\varepsilon)$,

- ***Low noise sensitivity:*** $\mathrm{NS}(p) \leq O(r + \varepsilon)$, *and*

- ***Low isolation probability:*** $\mathrm{iso}(p) \leq O(\varepsilon)$.

We achieve this with convex programming. The $\ell_1$ noise sensitivity is a convex constraint, and we use a convex upper bound on the $\ell_1$ analogue of the isolation probability. We minimize error over the set of degree-$\tilde{O}(1/\varepsilon^2)$ polynomials under these constraints. To show that a feasible polynomial of error $\mathsf{opt} + O(\varepsilon)$ exists, we show that the halfspace-approximating polynomial given in [DGJ+09] satisfies the constraints under any subgaussian isotropic log-concave distribution.

### 1.2.3 Randomized partitioning and rounding (Algorithm 5, Algorithm 4)

With $p$ in hand, we must then round $p$ to some polynomial threshold function $\mathrm{sign}(p - t)$ that satisfies the error, noise sensitivity, and isolation probability constraints. Simply rounding at a uniformly random threshold $t \sim [-1, 1]$, as in [KKMS08], does result in error $\mathsf{opt} + O(\varepsilon)$, noise sensitivity $O(r)$, and isolation probability $O(\varepsilon)$ in expectation, but doesn't guarantee that the conditions ever hold simultaneously for the same $t$. Thus, running the local corrector on just one rounded function would not guarantee a hypothesis with the right properties. In this step, we find a "deterministic mixture" of rounded functions that simultaneously satisfies the three conditions: a partition of the domain into parts, where each part is rounded at a different threshold. The corrector is run on each part separately.

A first attempt at finding such a partition would be to allow it to have $O(1/\varepsilon^2)$ parts. Observe that the error, noise sensitivity, and isolation probability concentrate with deviation $\varepsilon$ when averaging over $O(1/\varepsilon^2)$ random thresholds. Thus, an equal-weighted mixture of $O(1/\varepsilon^2)$ independent random roundings would satisfy all of the guarantees simultaneously with high probability. But suppose we partitioned the domain into $1/\varepsilon^2$ sets of equal mass — we can't guarantee robustness for any point near the boundaries of these sets, and the total volume of these boundaries scales inversely with $\varepsilon$, which is undesirable. To minimize the increase in boundary volume due to partitioning, it is necessary to find a partition of constant size.

We show that there exists a mixture of four rounded functions satisfying all the constraints simultaneously by applying Carathéodory's theorem ([Car07]), and we find the mixture by linear programming.

To understand how to turn the mixture into a partition, consider the example of handling just the error when $y$ is a deterministic function of $x$. The error of the mixture is

$$\sum_{i \in [4]} w_i \cdot \mathrm{err}(\mathrm{sign}(p - t_i)) \leq \mathsf{opt} + O(\varepsilon),$$

---

[3]We remark that in a setting where each call to the corrector is allowed to fail *independently* with probability $\delta$, the query complexity can be reduced from $d^{O(k)} \cdot \log(1/\delta)/\varepsilon^2$ to $O(\log(1/\delta)/\varepsilon^2)$ and the assumption that $f$ is a degree-$k$ PTF can be dropped. This is not applicable in our setting but might be desired in a non-learning application of local correction for robustness.

where $w_i$ is the mixing weight of the $i^{th}$ PTF. For each $i$ in the mixture, there is a set of volume $\mathrm{err}(\mathrm{sign}(p - t_i))$ consisting of the points misclassified by $\mathrm{sign}(p - t_i)$. We want to partition the domain such that a $w_i$-fraction of this set falls into the $i^{th}$ part, so that

$$\sum_{i \in [4]} \mathbb{1}[x \text{ is in the } i^{th} \text{ part and } y \neq \mathrm{sign}(p(x) - t_i)] = \sum_{i \in [4]} w_i \cdot \mathrm{err}(\mathrm{sign}(p - t_i)).$$

We cite a theorem of [DHV06], which says, essentially, that projecting on a random unit vector causes the set to be very close to normally-distributed with high probability. Thus, our partition is according to the inner product with a random unit vector $u$, and the parts are intervals $J_1, \ldots, J_4$ of Gaussian mass $w_1, \ldots, w_4$ respectively. With high probability, the sets of misclassified points and the sets for which robustness is guaranteed by the corrector are all partitioned with the correct weights.

**Theorem 1.4** (Randomized partitioning, informal version of Theorem 3.4). *Let $p$ be a polynomial satisfying the guarantees of Theorem 1.3. There is an algorithm that outputs a unit vector $u$, rounding thresholds $t_1, \ldots, t_4$, and a partition of the real line into intervals $J_1, \ldots J_4$ such that the hypothesis*

$$h(x) = \sum_{i=1}^{4} \textsc{RobustnessLCA}(\mathrm{sign}(p(x) - t_i)) \cdot \mathbb{1}[\langle x, u \rangle \in J_i]$$

*has the following properties with high probability:*

- ***Accuracy:*** $\mathrm{err}(h) \leq \mathsf{opt} + O(\varepsilon)$,

- ***Robustness:*** $\mathrm{Boundary}_{\mathcal{D}, r}(h) \leq O(r + \varepsilon)$.

*The time complexity is $d^{\tilde{O}(1/\varepsilon^2)}$.*

## 1.3 Verifiable robustness

The work of [GKVZ22] discusses the planting of "backdoors" in learned classifiers — structured violations of the robustness condition — and the impossibility of efficiently distinguishing a robust model from a backdoored one in the most general case. The takeaway is that when training is performed by an untrusted service, it is not generally possible to verify that a hypothesis is robust, even if a description of the hypothesis is available. However, our learning algorithm produces a hypothesis with a specific structure that can be checked, and the reduced expressivity of this structure allows robustness to be verifiable. Under complexity assumptions, the robustness of our algorithm's output can be verified in the following way:

**Corollary 1.5** (Deterministic verifiability, informal version of Corollary D.1). *If $\mathsf{P} = \mathsf{BPP}$, then there is a learning algorithm $\mathcal{B}$ that, given access to labeled samples from a subgaussian isotropic log-concave distribution, runs in time $d^{\tilde{O}(1/\varepsilon^2)}$ and produces a hypothesis $h$ with the following guarantees[4]:*

- ***Agnostic approximation:*** *$h$ satisfies $\Pr_{(x,y) \sim \mathcal{D}}[h(x) \neq y] \leq \mathsf{opt} + O(\varepsilon)$, where $\mathsf{opt}$ is the misclassification error of the best halfspace.*

- ***Verifiable robustness:*** *There is an efficient deterministic verifier that takes as input a hypothesis $g$ and a point $x \in \mathbb{R}^d$, and always rejects if*

$$\exists z : \|z\|_2 \leq r \text{ and } g(x) \neq g(x + z).$$

*If $g$ is the output of $\mathcal{B}$, then the verifier accepts with probability at least $1 - O(r + \varepsilon)$ over $x$ drawn from $\mathcal{D}$.*

As a result, training can be done entirely by an untrusted service who claims to be using our learning algorithm, but may not be. The verifier checks whether the hypothesis matches the format our learning algorithm produces, rejects if it doesn't match, and performs a robustness test that is sound if it does match. The user must trust that the error of the hypothesis really is $\mathsf{opt} + O(\varepsilon)$, but does not have to place any trust in the service to guarantee robustness — there is a deterministic algorithm that

---

[4]Agnostic approximation holds with high probability, whereas verifiable robustness always holds (deterministically).

the user can perform to verify that most points are not near the boundary. Furthermore, in a setting where the user performs some of the construction of the hypothesis (but need not access any training data), verifiable robustness can be made unconditional, but the soundness of the verifier can fail with small probability over the randomness of the hypothesis construction. See Appendix D for further discussion.

## 1.4 Related work.

**Adversarially robust learning.** Several recent papers study adversarially robust learning of various concept classes in the distribution-free PAC setting. These papers also generalize the set of adversarial perturbations beyond the $\ell_2$ ball. Some consider $\ell_p$ perturbations and some consider fully arbitrary sets.

The works of [CBM18, MHS19] show that there is essentially no statistical separation between robust learning and standard learning; thus any gap in the hardness of these tasks must be computational. The works of [BLPR19, DNV19, ADV19] exhibit concept classes and perturbation sets such that, under standard hardness assumptions, robust learning is computationally harder than standard learning. The work of [SHS+18] exhibits some concept classes and perturbation sets such that a high robust risk is inevitable.

The work of [MGDS20] gives an algorithm for robustly learning halfspaces in the *realizable* setting where the data is assumed to be labeled by a halfspace with random classification noise. The works of [DKM20, DKM19] give algorithms for robust, proper, agnostic learning of halfspaces; however, the data distributions are assumed to be supported on the unit ball. When scaled up to match the parameters of our setting — distributions concentrated on a radius-$\sqrt{d}$ ball — the running times of these algorithms have an exponential dependence on $\sqrt{d}$.

The work of [ADV19] also studies robust learning in the distribution-free setting. They give an algorithm for learning halfspaces and degree-2 PTFs with robustness under $\ell_\infty$-bounded perturbations.

In the distribution-specific setting, [GKKW21] gives an algorithm for learning monotone conjunctions with robustness to perturbations of $O(\log n)$ Hamming distance, under log-Lipschitz distributions on the Hamming cube. As mentioned earlier, the proper agnostic learner of [DKK+21] is a robust learner for halfspaces under the Gaussian distribution.

**Other work on learning halfspaces.** See Section 1 for comparison of our work with [DGJ+09, KKMS08]. Under general log-concave distributions, [ABL17] gives an algorithm for semiagnostic proper learning of origin-centered halfspaces under log-concave distributions, i.e. the algorithm gives a halfspace with error $O(\mathsf{opt}) + \varepsilon$, where $\mathsf{opt}$ is the error of best halfspace. In contrast to this, our work focuses on obtaining the optimal $\mathsf{opt} + \varepsilon$ error. [Dan15] gives a poly-time method of achieving error $(1 + \alpha)\mathsf{opt} + \varepsilon$ under for any constant $\alpha$, but via an improper learning algorithm that uses polynomial regression[5]. The work of [DKK+21], in addition to the $d^{\tilde{O}(1/\varepsilon^4)}$-time algorithm for achieving error $\mathsf{opt} + \varepsilon$ for halfspaces under the Gaussian distribution (discussed in Section 1), gives a poly-time proper learning algorithm for origin-centered halfspaces that achieves error $(1 + \alpha)\mathsf{opt} + \varepsilon$ (similar to [Dan15]). All algorithms in [DKK+21] are highly specific to Gaussian distributions and do not extend to, for example, to the uniform distribution over $[-1, 1]^d$ and other sub-Gaussian log-concave distributions.

## 2 Preliminaries

When dealing with a distribution $\mathcal{D}$ over $\mathbb{R}^d \times \{\pm 1\}$, we will sometimes overload the notation to use the symbol $\mathcal{D}$ to refer to the $\mathbb{R}^d$-marginal of $\mathcal{D}$. Analogously, when dealing with a collection $S$ of pairs $\{(x_i, y_i)\}$ in $\mathbb{R}^d \times \{\pm 1\}$ we will sometimes write $\Pr_{x \sim S}[\cdots]$ to refer to $\Pr_{(x,y) \sim S}[\cdots]$ when the values of $y$ are not referenced.

---

[5]The algorithm of [Dan15] also applies only to the Gaussian distribution, and not to general sub-Gaussian log-concave distributions.

## 2.1 Perturbation and robustness models

**Definition 2.1** ($r$-boundary volume)**.** *The boundary volume of a function $f$ at radius $r$ on distribution $\mathcal{D}$ is defined as*

$$\text{Boundary}_{\mathcal{D},r}(f) = \Pr_{x \sim \mathcal{D}}[\exists z : \|z\|_2 \leq r \text{ and } f(x) \neq f(x+z)].$$

**Definition 2.2** (Local noise sensitivity)**.** *We define the local noise sensitivity $\phi_{f,\eta}(x)$ at noise scale $\eta$ as*

$$\phi_{f,\eta}(x) := \mathbb{E}_{z \sim \mathcal{N}(0,I_d)}[\tfrac{1}{2}|f(x) - f(x+\eta z)|].$$

*We remark that when $f$ is Boolean-valued, this is equivalently*

$$\phi_{f,\eta}(x) := \Pr_{z \sim \mathcal{N}(0,I_d)}[f(x) \neq f(x+\eta z)].$$

**Definition 2.3** (Distributional noise sensitivity)**.** *The noise sensitivity of a function $f$ at noise scale $\eta$ on distribution $\mathcal{D}$ is defined as*

$$\text{NS}_{\mathcal{D},\eta}(f) = \mathbb{E}_{x \sim \mathcal{D}}[\phi_{f,\eta}(x)].$$

**Definition 2.4** (Isolation probability)**.** *We call a point* isolated *if its local noise sensitivity is over a threshold. The* isolation probability *of a function $f$ relative to a threshold $t$ is*

$$\text{iso}_{\mathcal{D},\eta}(f,t) := \Pr_{x \sim \mathcal{D}}[\phi_{f,\eta}(x) > t].$$

*We also use a convex relaxation of the isolation indicator and define $\psi$ to be its expectation:*

$$\psi_{\mathcal{D},\eta}(f) := 10 \mathbb{E}_{x \sim \mathcal{D}}[\mathbb{1}[\phi_{f,\eta}(x) > 0.6] \cdot (\phi_{f,\eta}(x) - 0.6)].$$

*Note that $\psi(f)$ is an upper bound on $\text{iso}(f, 0.7)$.*

### 2.1.1 Noise sensitivity approximators

Below, we introduce the notion of a local noise sensitivity approximator. Many of the algorithms we introduce will use such an approximator as a black box they can query (see Algorithm 1, Algorithm 5, Algorithm 4 and Algorithm 3). We first introduce a local noise sensitivity approximator and some related notions, and present a randomized method for efficiently instantiating it.[6]

**Definition 2.5** (Local noise sensitivity approximator for PTFs)**.** *For a degree parameter $k$, an algorithm $\hat{\phi}$ is an $\varepsilon$-accurate noise sensitivity approximator if whenever it is given (i) a degree-$k$ polynomial threshold function $f$, (ii) a noise rate $\eta \in (0,1)$, and (iii) a point $x \in \mathbb{R}^d$, it outputs a value $\hat{\phi}_{f,\eta}(x)$ such that*

$$\left| \hat{\phi}_{f,\eta}(x) - \phi_{f,\eta}(x) \right| \leq \varepsilon.$$

**Definition 2.6.** *Let $\hat{\phi}$ be as in the definition above. Then we define the empirical noise sensitivity and isolation probabilities as*

$$\widehat{\text{NS}}_{\mathcal{D},\eta}(f) := \mathbb{E}_{x \sim \mathcal{D}} \hat{\phi}_{f,\eta}(x) \quad and \quad \widehat{\text{iso}}_{\mathcal{D},\eta}(f,t) := \mathbb{E}_{x \sim \mathcal{D}} \mathbb{1}[\hat{\phi}_{f,\eta}(x) > t].$$

*We also define*

$$\hat{\psi}_{\mathcal{D},\eta}(f) = 10 \mathbb{E}_{x \sim \mathcal{D}}[\mathbb{1}[\hat{\phi}_{f,T,\eta}(x) > 0.6] \cdot (\phi_{f,\eta}(x) - 0.6)].$$

## 2.2 Distances and errors

**Definition 2.7** (Error and optimal error)**.** *We will denote the error of a hypothesis on a distribution*

$$\text{err}_{\mathcal{D}}(h) := \mathbb{E}_{(x,y) \sim \mathcal{D}}\left[\tfrac{1}{2}|h(x) - y|\right]$$

*and remark that when $h$ is Boolean-valued this is equivalently*

$$\text{err}_{\mathcal{D}}(h) := \Pr_{(x,y) \sim \mathcal{D}}[h(x) \neq y].$$

---

[6]In Appendix A.1 a randomized construction of $\hat{\phi}$ is given and in Appendix D a deterministic method is presented assuming BPP = P.

*Relative to a sample set we use*

$$\widehat{\mathrm{err}}_S(h) = \frac{1}{|S|} \sum_{(x,y) \in S} \tfrac{1}{2}|h(x) - y|.$$

*We will often denote by* opt *the optimal error of a function $f$ with respect to a concept class $\mathcal{F}$:*

$$\mathsf{opt} := \inf_{g \in \mathcal{F}} \mathrm{err}_{\mathcal{D}}(g).$$

*In this work, $\mathcal{F}$ is always the class of linear threshold functions over $\mathbb{R}^d$.*

## 2.3 Miscellaneous

We will use the notation of the form $a = b \pm c$ as a shorthand for $|a - b| \leq c$. We will also drop subscripts, particularly on $\phi$ and $\hat{\phi}$, when they are clear from context. Throughout this work, wherever there is noise, the noise scale $\eta$ is always $10r$, where $r$ is the desired robustness radius. We use the notation $t \sim [-1, 1]$ to denote that $t$ is drawn uniformly from $[-1, 1]$.

## 3  Results and pseudocode of our main algorithm

Our main result is the following. It states the correctness and running time of the main algorithm, ROBUSTLEARN.

**Theorem 3.1** (Correctness and complexity of Algorithm 1)**.** *Let $\varepsilon \geq d^{-1/7}$, let $r, \delta \in (0, 1)$, and let $C$ be a sufficiently large absolute constant. Furthermore, let $\hat{\phi}$ be an $\varepsilon$-accurate local noise sensitivity approximator for degree-$k$ PTFs over $\mathbb{R}^d$, where $k := \frac{C \log^2(1/\varepsilon)}{\varepsilon^2}$, and let the running time and query complexity of $\hat{\phi}$ be $\tau$.*

*Let $\mathcal{F}$ be the class of linear threshold functions over $\mathbb{R}^d$ and $\mathcal{D}$ be a distribution over $\mathbb{R}^d \times \{-1, 1\}$ whose $\mathbb{R}^d$-marginal is isotropic, subgaussian and log-concave. The algorithm ROBUSTLEARN, given i.i.d. sample access to the distribution $\mathcal{D}$, with probability at least $1 - O(\delta)$ returns a hypothesis $h : \mathbb{R}^d \to \{\pm 1\}$ for which the following hold:*

$$\mathrm{err}_{\mathcal{D}}(h) \leq \mathsf{opt} + O(\varepsilon)$$

$$\mathrm{Boundary}_{\mathcal{D},r}(h) \leq O(r + \varepsilon).$$

*The running time and sample complexity are* $\mathrm{poly}\left(d^{\log^2(1/\varepsilon)/\varepsilon^2} \cdot \log(1/\delta) \cdot \tau\right)$.

**Corollary 3.2.** *By instantiating $\hat{\phi}$ with the algorithm of Appendix A.1, which evaluates queries in* $\mathrm{poly}\left(d^{\log^2(1/\varepsilon)/\varepsilon^2} \cdot \log(1/\delta)\right)$ *time, ROBUSTLEARN runs in total time* $\mathrm{poly}\left(d^{\log^2(1/\varepsilon)/\varepsilon^2} \cdot \log(1/\delta)\right)$.

---

**Algorithm 1** ROBUSTLEARN($\mathcal{D}, \varepsilon, \delta, r$):

---

1: **Input:** sample access to distribution $\mathcal{D}$ over $\mathbb{R}^d \times \{\pm 1\}$, error bound $\varepsilon$,
2:      confidence bound $\delta$, robustness radius $r$
3: **Uses:** local noise sensitivity approximator $\hat{\phi}$.                    (See Definition 2.5)
4: **Output:** hypothesis $h : \mathbb{R}^d \to \{-1, 1\}$
5: $p := $ LEARNREALVALUED($\mathcal{D}, \varepsilon, r$).                    (See Algorithm 2)
6: **return** COMPUTECLASSIFIER($\mathcal{D}, p, r, \varepsilon, \delta$).                    (See Algorithm 5)

---

The algorithm has two "phases": LEARNREALVALUED, which solves a convex relaxation of the learning task, and COMPUTECLASSIFIER, which rounds the real-valued hypothesis to a Boolean one with small boundary volume. We present and analyze these algorithms in Appendix B.3 and Appendix C.3 respectively.

**Theorem 3.3** (Correctness of Algorithm 2, LEARNREALVALUED)**.** *Let $r \in (0,1)$, and let $\mathcal{D}$ be a distribution over $\mathbb{R}^d \times \{\pm 1\}$ whose $\mathbb{R}^d$-marginal is a subgaussian isotropic log-concave distribution over $\mathbb{R}^d$. Let $\mathcal{F}$ be the class of halfspaces on $\mathbb{R}^d$. Then, for some sufficiently large absolute constant $C$, the algorithm LearnRealValued, given $\varepsilon, \delta$ and sample access to $\mathcal{D}$, runs in time $\mathrm{poly}\left(d^{O(\log^2(1/\varepsilon)/\varepsilon^2)}\right)$ and will with probability at least $1 - O(\delta)$ return a degree-$O(\log^2(1/\varepsilon)/\varepsilon^2)$ polynomial $p$ for which the following hold:*

$$\underset{t \sim [-1,1]}{\mathbb{E}}\left[\underset{(x,y) \sim D}{\mathbb{E}}\left[\phi_{\mathrm{sign}(p-t),10r}(x)\right]\right] \le 100r + O(\varepsilon), \tag{2}$$

$$\underset{(x,y) \sim \mathcal{D}}{\mathrm{Pr}}\left[\underset{t \sim [-1,1]}{\mathbb{E}}[\phi_{\mathrm{sign}(p(x)-t),10r}(x)] \ge 0.7\right] \le O(\varepsilon), \tag{3}$$

$$\underset{t \sim [-1,1]}{\mathbb{E}}\left[\underset{(x,y) \sim D}{\mathrm{Pr}}\left[\mathrm{sign}(p(x) - t) \ne y\right]\right] \le \mathsf{opt} + O(\varepsilon). \tag{4}$$

**Theorem 3.4** (Correctness of Algorithm 5, COMPUTECLASSIFIER)**.** *Let $\varepsilon \ge d^{-1/7}$ and $\delta, r > 0$. Let $\hat{\phi}$ be an $\varepsilon$-accurate local noise sensitivity approximator for degree-$k$ PTFs over $\mathbb{R}^d$. Let $\mathcal{D}$ be a distribution over $\mathbb{R}^d \times \{-1, 1\}$ such that the $\mathbb{R}^d$-marginal is isotropic and log-concave. Let $p : \mathbb{R}^d \to \mathbb{R}$ be a polynomial of degree at most $k$ satisfying*

$$\underset{t \sim [-1,1]}{\mathbb{E}}\left[\underset{(x,y) \sim \mathcal{D}}{\mathbb{E}}[\phi_{\mathrm{sign}(p-t),10r}(x)]\right] \le 100r + O(\varepsilon), \tag{5}$$

$$\underset{(x,y) \sim \mathcal{D}}{\mathrm{Pr}}\left[\underset{t \sim [-1,1]}{\mathbb{E}}[\phi_{\mathrm{sign}(p(x)-t),10r}(x)] \ge 0.7\right] \le O(\varepsilon). \tag{6}$$

*Then the algorithm* COMPUTECLASSIFIER$(\mathcal{D}, p, r, \varepsilon, \delta)$*, given sample access to $\mathcal{D}$ and query access to $\hat{\phi}$, outputs a hypothesis $h : \mathbb{R}^d \to \{-1, 1\}$ such that the following properties hold with probability at least $1 - O(\delta)$:*

$$\mathrm{err}_{\mathcal{D}}(h) \le \underset{t \sim [-1,1]}{\mathbb{E}}[\mathrm{err}_{\mathcal{D}}(\mathrm{sign}(p-t))] + O(\varepsilon)$$

$$\mathrm{Boundary}_{\mathcal{D},r}(h) \le O(r + \varepsilon).$$

*The running time and number of queries to $\hat{\phi}$ are $\mathrm{poly}(d^k \cdot 1/\varepsilon \cdot \log(1/\delta))$.*

## 3.1 Proof of Theorem 3.1.

In this section we prove the main theorem assuming Theorem 3.4 and Theorem 3.3.

*Proof of Theorem 3.1.* By Theorem 3.3, with probability $1 - O(\delta)$, the output $p$ of LEARNREAL-VALUED is a degree-$k$ polynomial satisfying

$$\underset{t \sim [-1,1]}{\mathbb{E}}\left[\underset{(x,y) \sim D}{\mathbb{E}}\left[\phi_{\mathrm{sign}(p-t),10r}(x)\right]\right] \le 100r + O(\varepsilon), \tag{7}$$

$$\underset{(x,y) \sim \mathcal{D}}{\mathrm{Pr}}\left[\underset{t \sim [-1,1]}{\mathbb{E}}[\phi_{\mathrm{sign}(p(x)-t),10r}(x)] \ge 0.7\right] \le O(\varepsilon), \tag{8}$$

$$\underset{t \sim [-1,1]}{\mathbb{E}}\left[\mathrm{Pr}_{(x,y) \sim D}\left[\mathrm{sign}(p(x) - t) \ne y\right]\right] \le \mathsf{opt} + O(\varepsilon). \tag{9}$$

This polynomial $p$ is the input to COMPUTECLASSIFIER. We see that the premises of Theorem 3.4 are satisfied by the guarantees of Equation (7) and Equation (8). Taking the conclusions of Theorem 3.4 and combining it with Equation (9) we see that the hypothesis $h$ we return satisfies

$$\mathrm{err}_{\mathcal{D}}(h) \le \underset{t \sim [-1,1]}{\mathbb{E}}[\mathrm{err}_{\mathcal{D}}(\mathrm{sign}(p-t))] + O(\varepsilon) \le \mathsf{opt} + O(\varepsilon),$$

$$\mathrm{Boundary}_{\mathcal{D},r}(h) \le O(r + \varepsilon),$$

which concludes the proof of correctness.

We now analyze the sample complexity and running time. By [Theorem 3.3](#), LEARNREALVALUED takes $d^{O(\log^2(1/\varepsilon)/\varepsilon^2)}\operatorname{poly}(\log(1/\delta))$ time and samples. By [Theorem 3.4](#), since the degree of $p$ is $O(\log^2(1/\varepsilon)/\varepsilon^2)$ and $\varepsilon \geq d^{-1/7}$, COMPUTECLASSIFIER takes $d^{O(\log^2(1/\varepsilon)/\varepsilon^2)}\operatorname{poly}(\log(1/\delta))$ time and queries to $\hat{\phi}$. Evaluations of $\hat{\phi}$ take time $\tau$, so the total running time is

$$\operatorname{poly}(d^{\log^2(1/\varepsilon)/\varepsilon^2} \cdot \log(1/\delta) \cdot \tau),$$

and all components succeed with probability $\geq 1 - O(\delta)$. $\qquad\square$

## Acknowledgments

We thank Ronitt Rubinfeld for proving insightful comments on a draft of this paper. We are also thankful to Vinod Vaikuntanathan for helpful discussions regarding backdoors and verifiable robustness.

Jane Lange is supported by NSF Awards CCF-2006664 and CCF-2310818, and NSF Graduate Research Fellowships Program, and Arsen Vasilyan is supported in part by NSF AI Institute for Foundations of Machine Learning (IFML), NSF awards CCF-2006664, CCF-2310818 and Big George Fellowship. Part of this work was conducted while the authors were visiting the Simons Institute for the Theory of Computing.

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

## A  Further preliminaries

### A.1  Randomized implementation of $\hat{\phi}$

Let $T$ be a sufficiently large set of i.i.d. samples from $\mathcal{N}(0, I_d)$. Below, we show that the choice of $\hat{\phi}$ as the following empirical estimate will satisfy Definition 2.5 with high probability over the choice of $T$:

$$\tilde{\phi}_{f,T,\eta}(x) = \frac{1}{|T|} \sum_{z \in T} \frac{|f(x) - f(x + \eta z)|}{2}.$$

**Fact A.1** ([Ant95]). *The VC dimension of the class of polynomial threshold functions of degree $k$ is* $\Theta(\binom{d}{\leq k}) = O(d^k)$.

**Fact A.2** (Concentration of expectations from VC dimension). *Let $\mathcal{C}$ be a concept class and $\mathcal{D}$ be a distribution over $\mathbb{R}^d$. For some sufficiently large absolute constant $C$, the following is true. With probability at least $1 - \delta$ over a sample $S$ of i.i.d. samples from $\mathcal{D}$, with $|S| \geq C \cdot VC(\mathcal{C}) \log(1/\delta)/\varepsilon^2$, the following holds:*

$$\sup_{f \in \mathcal{C}} \left| \mathbb{E}_{x \sim \mathcal{D}}[f(x)] - \frac{1}{|T|} \sum_{x \in T} f(x) \right| \leq \varepsilon.$$

We will call a sample set "representative" of distribution $\mathcal{D}$ for a concept class if it satisfies the above success condition.

**Claim A.3** (Representative samples provide accurate noise sensitivity estimates). *Let $\mathcal{C}$ be the class of degree-$k$ PTFs. Let $T$ be a set of points in $\mathbb{R}^d$ that is representative for $\mathcal{N}(0, I_d)$ for $\mathcal{C}$ (in the sense of Fact A.2). Then with probability $\geq 1 - \delta$ over $T$, the following holds for all $f \in \mathcal{C}$, $x \in \mathbb{R}^d$, and $\eta \in (0, 1)$:*

$$\left| \tilde{\phi}_{f,T,\eta}(x) - \phi_{f,\eta}(x) \right| \leq \varepsilon.$$

*Proof.* Observe that for any fixed $x$, the function $g(z) = -f(x) \cdot f(x + \eta z)$ is a member of $\mathcal{C}$. Thus by Fact A.2, we have with probability $1 - \delta$ that

$$\left| \mathbb{E}_{z \sim T}[g(z)] - \mathbb{E}_{z \sim \mathcal{N}(0, I_d)}[g(z)] \right| \leq \varepsilon.$$

The claim follows from observing that $\phi_{f,r}(x) = \frac{1}{2}(\mathbb{E}[g(z)] + 1)$, and thus

$$\left| \tilde{\phi}_{f,T,\eta}(x) - \phi_{f,\eta}(x) \right| = \frac{1}{2} \left| \mathbb{E}_{z \sim T}[g(z)] - \mathbb{E}_{z \sim \mathcal{N}(0, I_d)}[g(z)] \right| \leq \varepsilon.$$

$\square$

## A.2 Distances and errors

The following facts about the error and local noise sensitivity after rounding a real-valued function to Boolean follow directly from linearity of expectation.

**Fact A.4** (Randomized rounding). *Let $f$ and $g$ be real-valued functions. Let the rounded function $f_t$ be defined as*

$$f_t(x) = \text{sign}(f(x) - t),$$

*and likewise for $g_t$. Then,*

$$\mathbb{E}_{t \sim [-1,1]} \left[ \Pr_{(x,y) \sim \mathcal{D}}[f_t(x) \neq y] \right] \leq \mathbb{E}_{(x,y) \sim \mathcal{D}} \left[ \frac{|f(x) - y|}{2} \right].$$

$$\mathbb{E}_{t \sim [-1,1]} \left[ \Pr_{(x,y) \sim \mathcal{D}}[f_t(x) \neq g_t(x)] \right] \leq \mathbb{E}_{(x,y) \sim \mathcal{D}} \left[ \frac{|f(x) - g(x)|}{2} \right].$$

*Similarly,*[7]

$$\mathbb{E}_{t \sim [-1,1]}[\phi_{f_t,\eta}(x) \leq \phi_{f,\eta}(x) \quad \text{and} \quad \mathbb{E}_{t \sim [-1,1]}[\tilde{\phi}_{f_t,T,\eta}(x) \leq \tilde{\phi}_{f,T,\eta}(x)$$

*for any $x$, $\eta$ and $T$.*

Finally, the following is a standard fact, see for example [DMR18].

**Fact A.5** (TV distance between one-dimensional Gaussians). *For all $\mu_1, \mu_2$ in $\mathbb{R}$ and $\sigma_1, \sigma_2$ in $\mathbb{R}_{>0}$ we have*

$$d_{\text{TV}}(\mathcal{N}(\mu_1, \sigma_1^2), \mathcal{N}(\mu_2, \sigma_2^2)) \leq \frac{3|\sigma_1^2 - \sigma_2^2|}{2\sigma_1^2} + \frac{|\mu_1 - \mu_2|}{2\sigma_1}$$

---

[7]See Appendix A.1 for the definition of $\tilde{\phi}$.

### A.3 Miscellaneous

We say that $f$ is a *randomized Boolean function* over $\mathbb{R}^d$ if $f$ maps every point in $\mathbb{R}^d$ to a random variable $f(x)$ over $\{0,1\}$. Sometimes we will write $f(x) = \mathrm{Ber}(p)$, to denote that $f(x) = 1$ with probability $p$ and $0$ with probability $1 - p$. In Appendix E, miscellaneous claims about uniform convergence are shown, which are used in the analysis of Algorithm 2.

## B Pseudocode and analysis of LearnRealValued

In this section we present and analyze our algorithm for learning a polynomial with small noise sensitivity and isolation probability.

---

**Algorithm 2** LEARNREALVALUED$(D, \varepsilon, r)$:

---

1: **Input:** Sample access to distribution $D$, error bound $\varepsilon$, robustness radius $r$
2: **Output:** hypothesis $h : \{-1, 1\}^d \to \mathbb{R}$
3: Set $k := \frac{C \log^2(1/\varepsilon)}{\varepsilon^2}$.
4: **for** $i$ in $\{1, \cdots, \log(1/\delta)\}$ **do**
5:      $S_i := \{Cd^{Ck} \cdot \log(1/\delta)/\varepsilon^2$ i.i.d. draws from $D\}$.
6:      $T_i := \{Cd^{Ck} \cdot \log(1/\delta)/\varepsilon^2$ i.i.d. draws from $\mathcal{N}(0, I_d)\}$
7:      Let $p_i$ be the output of the following convex program:
- Domain: polynomials over $\mathbb{R}^d$ of degree $k$.
- Minimize: $\sum_{(x,y) \in S} |p(x) - y|$
- Constraints:
    - Define $\tilde{\phi}(x)_{p,T,10r} := \frac{1}{|T|} \sum_{z \in T} \frac{|f(x) - f(x + \eta z)|}{2}$
    - Define $\tilde{\psi}_{p,T,10r} := 10 \cdot \mathbb{1}[\tilde{\phi}(x)_{p,T,10r} > 0.6] \cdot (\tilde{\phi}(x)_{p,T,10r} - 0.6)$
    - Constraint 1: $\frac{1}{|S|} \sum_{x \in S} \tilde{\phi}_{p,T,10r}(x) \leq 100r + \varepsilon$
    - Constraint 2: $\frac{1}{|S|} \sum_{x \in S} \tilde{\psi}_{p,T,10r}(x) \leq \varepsilon$

8: $i^* := \operatorname{argmin}_i \left( \mathbb{E}_{(x,y) \sim S_i} \left[ \frac{|p_i(x) - y|}{2} \right] \right)$
9: $p := p_{i^*}$.
10: **return** $p$

---

### B.1 Facts about log-concave distributions.

**Definition B.1.** *A distribution $D$ over $\mathbb{R}^d$ is is called $C$-sub-Gaussian, if for any unit vector $u$:*

$$\Pr_{x \sim D}[|u \cdot x| > t] \leq \exp(-Ct^2)$$

*for any $t \geq 0$. We say $D$ is sub-Gaussian, if $D$ is $C$-sub-Gaussian for some absolute constant $C$.*

See e.g. [SW14] for the following fact:

**Fact B.1.** *If $D_1$ and $D_2$ are log-concave distributions over $\mathbb{R}^d$, then the convolution of $D_1$ and $D_2$ is also a log-concave distribution over $\mathbb{R}^d$.*

Likewise, the following fact follows directly from the definition of a sub-Gaussian distribution:

**Fact B.2.** *If $D_1$ and $D_2$ are sub-Gaussian distributions over $\mathbb{R}^d$, then the convolution of $D_1$ and $D_2$ is also a sub-Gaussian distribution over $\mathbb{R}^d$.*

The following fact can be found in [SW14]:

**Fact B.3.** *If a distribution $D$ over $\mathbb{R}^d$ is isotropic log-concave, then the projection of $D$ on any linear subspace in $\mathbb{R}^d$ is likewise log-concave.*

The following two facts can be found in [GKK23], [ABL17] and the references therein.

**Fact B.4.** *If a distribution $D$ over $\mathbb{R}^d$ is isotropic log-concave, then for some absolute constant $C_3$, for any unit vector $u$ and every interval $J \subset \mathbb{R}$, we have $\Pr_{x \sim D}[\langle u, x \rangle \in J] \leq |J|$.*

**Remark B.5.** It follows from a direct change-of-basis argument that Fact B.4 still holds if the premise that the distribution $D$ is isotropic[8] is replaced by a slightly more general premise that $D$ mean zero and satisfies:

$$I_{d \times d} \preceq \mathbb{E}_{x \sim D}[xx^T] \preceq 200 I_{d \times d}$$

For the following fact, see e.g. [DKS18] and the references therein:

**Fact B.6.** *If a distribution $D$ over $\mathbb{R}^d$ is log-concave, then for some absolute constant $C_4$, if $p$ is a degree-$k$ polynomial for which $\mathbb{E}_{x \sim D}[(p(x))^2] \leq 1$, then for any $B > e^k$ it is the case that*

$$\Pr_{x \sim D}[(p(x))^2 > B] \leq \exp(-C_4 B^{1/k})$$

One can use the fact above to conclude the following:

**Observation B.7.** *If a distribution $D$ over $\mathbb{R}^d$ is log-concave, then for some absolute constant $C_5$, for every degree-$k$ polynomial $p$ it is the case that*

$$\mathbb{E}_{x \sim D}[|p(x)|] \geq \frac{\sqrt{\mathbb{E}_{x \sim D}[(p(x))^2]}}{(C_5 k)^{C_5 k}}$$

*Proof.* Without loss of generality, we can assume that $\mathbb{E}_{x \sim D}[(p(x))^2] = 1$. For any $B \geq 0$, we have

$$\mathbb{E}_{x \sim D}[|p(x)|] \geq \frac{\mathbb{E}_{x \sim D}[|p(x)|^2 \, \mathbb{1}_{|p(x)| \leq B}]}{B} = \frac{1 - \mathbb{E}_{x \sim D}[|p(x)|^2 \, \mathbb{1}_{|p(x)| > B}]}{B} \tag{10}$$

Using Fact B.6, we have

$$\mathbb{E}_{x \sim D}[|p(x)|^3] \leq e^{O(k)} + \int_{\beta=0}^{+\infty} \Pr_{x \sim D}[|p(x)|^3 > \beta] d\beta \leq$$

$$e^{O(k)} + \int_{\beta'=0}^{+\infty} (\beta')^{3k/2-1} \exp(-C_4 \beta') d\beta' \leq (O(k))^{O(k)},$$

which allows us to conclude that

$$\mathbb{E}_{x \sim D}[|p(x)|^2 \, \mathbb{1}_{|p(x)| > B}] \leq \frac{\mathbb{E}_{x \sim D}[|p(x)|^3]}{B} = \frac{(O(k))^{O(k)}}{B}.$$

Taking $B$ to equal $(C'k)^{C'k}$ for a sufficiently large absolute constant $C'$, and substituting the bound above into Equation (10) finishes the proof. $\quad\square$

## B.2 Approximating halfspaces with polynomials

**Claim B.8.** *Let $h$ be a linear threshold function over $\mathbb{R}^d$. Then, for every absolute constant $C$, there exists a polynomial $p$ of degree $O(1/\varepsilon^2 \log^2(1/\varepsilon))$, such that for any $C$-sub-Gaussian log-concave distribution $D$ over $\mathbb{R}^d$ satisfying $I_{d \times d} \preceq \mathbb{E}_{x \sim D}[xx^T] \preceq 200 I_{d \times d}$ we have*

$$\mathbb{E}_{x \sim D}[|h(x) - p(x)|] \leq O(\varepsilon)$$

**Remark B.9.** Recall that a distribution $D$ is log-concave if the logarithm of the probability density function of $D$ is concave. Note that the constants hidden by the $O(\cdot)$ notation in Claim B.8 are allowed to depend on the sub-Gaussianity constant $C$.

To prove the claim above, we will need the following proposition from [DGJ+09]:

**Fact B.10** ([DGJ+09]). *There exist absolute constants $C_4$ and $C_5$, such that for every $\varepsilon \in (0, 0.1)$, there exists a univariate polynomial $P$ satisfying the following. Denoting $a := \frac{\varepsilon^2}{C_4 \log(1/\varepsilon)}$ and $K := C_5 \frac{\log \frac{1}{\varepsilon}}{a} = O\left(\frac{\log^2 1/\varepsilon}{\varepsilon^2}\right)$, it is the case that*

- *The degree of $P$ is at most $K$.*

- *$P(t) \in [\text{sign}(t), \text{sign}(t) + \varepsilon]$ for all $t$ in $[-1/2, -2a] \cup [0, 1/2]$.*

---

[8]Recall that the condition that $D$ is isotropic means that $D$ has zero mean and covariance $I_{d \times d}$.

- $P(t) \in [-1, 1 + \varepsilon]$ *for t in* $[-2a, 0]$
- $|P(t)| \leq 2 \cdot (4t)^K$ *for all* $|t| \geq 1/2$.

The proof below follows closely the line of reasoning in [DGJ$^+$09] and is included for completeness.

*Proof of Claim B.8.* For a unit vector $u$ $h(x) = \text{sign}(u \cdot x - \tau)$ (assuming $\tau = 0$ for now)

If $|\tau| > \log(1/\varepsilon)$ the statement follows immediately by taking $P$ to be the constant polynomial that equals to $\text{sign}(\tau)$ everywhere in $\mathbb{R}^d$. The error between $h$ and $P$ will be upper-bounded by $O(\varepsilon)$ via Definition B.1 and Remark B.5.

Otherwise, we have $|\tau| \leq \log(1/\varepsilon)$. Let $P$ be the polynomial in Fact B.10 and take $w := C_6 \left( \frac{\log^2 1/\varepsilon}{\varepsilon^2} \right)^{1/2}$ for a sufficiently large absolute constant $C_6$. Denoting by $D_{\text{projected}}$ the projection of distribution $D$ on the unit vector $u$, we can bound the error as follows using Fact B.10:

$$\mathbb{E}_{x \sim D}[|\text{sign}(u \cdot x - \tau) - P((u \cdot x - \tau)/w)|] = \mathbb{E}_{z \sim D_{\text{projected}}}[|\text{sign}(z - \tau) - P((z - \tau)/w)|] \leq$$

$$\varepsilon + 2 \Pr_{z \sim D_{\text{projected}}}[-2a \leq z/w - \tau \leq 0] + \mathbb{E}_{z \sim D_{\text{projected}}}\left[ \left( \left( 2 \cdot \left( 4 \cdot \frac{z - \tau}{w} \right)^K + 1 \right) \mathbb{1}_{|(z-\tau)/w| \geq 1/2} \right) \right]$$

$$(11)$$

The second term above is bounded by $O(aw)$ due to Fact B.4 and Remark B.5. We also note that since $|\tau| \leq \log(1/\varepsilon)$ and $w := C_6 \left( \frac{\log^2 1/\varepsilon}{\varepsilon^2} \right)^{1/2}$, for sufficiently large absolute constant $C_6$ we gave $|\tau|/w \leq 0.1$. Therefore, whenever $(z - \tau)/w \geq 1/2$ we also have $z \geq 0.4w$. This allows us to upper bound

$$\mathbb{E}_{x \sim D}[|\text{sign}(u \cdot x - \tau) - P((u \cdot x - \tau)/w)|] \leq$$

$$\varepsilon + O(aw) + \mathbb{E}_{z \sim D_{\text{projected}}}\left[ \left( 2 \cdot \left( 4 \cdot \frac{z}{w} + 0.1 \right)^K + 1 \right) \mathbb{1}_{|z| \geq 0.4w} \right] \leq$$

$$\varepsilon + O(aw) + \mathbb{E}_{z \sim D_{\text{projected}}}\left[ \left( 3 \cdot \left( 5 \cdot \frac{z}{w} \right)^K \right) \mathbb{1}_{|z| \geq 0.4w} \right] \quad (12)$$

Decomposing the interval $[0.4, \infty]$ into a union $\bigcup_{i=0}^{\infty}[0.4 \cdot 2^i, 0.4 \cdot 2^{i+1}]$ and using Definition B.1 and Remark B.5 we can now upper-bound the last term above as follows:

$$\mathbb{E}_{z \sim D_{\text{projected}}}\left[ \left( 3 \cdot \left( 5 \cdot \frac{z}{w} \right)^K \right) \mathbb{1}_{|z| \geq 0.4w} \right] \leq$$

$$2 \sum_{i=0}^{\infty} \left( 3 \cdot (4 \cdot 2^i)^K \Pr_{z \sim D_{\text{projected}}}\left[ |z| \geq 2^{i-1} \cdot 0.4w \right] \right) \leq$$

$$6 \sum_{i=0}^{\infty} \left( (4 \cdot 2^i)^K \exp(-C(2^{i-3}w)^2) \right) \quad (13)$$

Substituting $K = O\left( \frac{\log^2 1/\varepsilon}{\varepsilon^2} \right)$ and $w = C_6 \left( \frac{\log^2 1/\varepsilon}{\varepsilon^2} \right)^{1/2}$, we can bound each term in the sum above

$$(4 \cdot 2^i)^K \exp(-C(2^{i-3}w)^2) \leq \exp\left( (i+2) \cdot O\left( \frac{\log^2 1/\varepsilon}{\varepsilon^2} \right) - C \cdot C_6 2^{i-3} \frac{\log^2 1/\varepsilon}{\varepsilon^2} \right)$$

For a sufficiently large absolute constant $C_6$ the above is at most $\varepsilon/2^{i+1}$, which substituted into Equation (13) gives us

$$\mathbb{E}_{z \sim D_{\text{projected}}}\left[ \left( 3 \cdot \left( 5 \cdot \frac{z}{w} \right)^K \right) \mathbb{1}_{|z| \geq 0.4w} \right] \leq \varepsilon.$$

Finally, substituting the inequality above into Equation (12), we get

$$\mathbb{E}_{x \sim D}[|\text{sign}(u \cdot x) - P(u \cdot x/w)|] = O(\varepsilon),$$

finishing the proof. $\square$

## B.3 Analyzing LearnRealValued.

**Theorem 3.3** (Correctness of Algorithm 2, LEARNREALVALUED)*. Let $r \in (0, 1)$, and let $\mathcal{D}$ be a distribution over $\mathbb{R}^d \times \{\pm 1\}$ whose $\mathbb{R}^d$-marginal is a subgaussian isotropic log-concave distribution over $\mathbb{R}^d$. Let $\mathcal{F}$ be the class of halfspaces on $\mathbb{R}^d$. Then, for some sufficiently large absolute constant $C$, the algorithm LearnRealValued, given $\varepsilon, \delta$ and sample access to $\mathcal{D}$, runs in time* $\text{poly}\left(d^{O(\log^2(1/\varepsilon)/\varepsilon^2)}\right)$ *and will with probability at least $1 - O(\delta)$ return a degree-$O(\log^2(1/\varepsilon)/\varepsilon^2)$ polynomial $p$ for which the following hold:*

$$\mathop{\mathbb{E}}_{t \sim [-1,1]} \left[ \mathop{\mathbb{E}}_{(x,y) \sim D} \left[ \phi_{\text{sign}(p-t),10r}(x) \right] \right] \leq 100r + O(\varepsilon), \tag{2}$$

$$\Pr_{(x,y) \sim \mathcal{D}} \left[ \mathop{\mathbb{E}}_{t \sim [-1,1]} [\phi_{\text{sign}(p(x)-t),10r}(x)] \geq 0.7 \right] \leq O(\varepsilon), \tag{3}$$

$$\mathop{\mathbb{E}}_{t \sim [-1,1]} \left[ \Pr_{(x,y) \sim D} \left[ \text{sign}(p(x) - t) \neq y \right] \right] \leq \mathsf{opt} + O(\varepsilon). \tag{4}$$

We first argue that the run-time is indeed $d^{O(\log^2(1\varepsilon)/\varepsilon^2)}$. The algorithm operates with polynomials of degree $k = O(\log^2(1\varepsilon)/\varepsilon^2)$ over $\mathbb{R}^d$, which are described via $d^O(k)$ coefficients. It remains to show that the constraints for the optimization task are indeed convex in the coefficients in the polynomial $p$. Recall that the constraints are:

- Constraint 1: $\frac{1}{|S|} \sum_{x \in S} \tilde{\phi}_{p,T,10r}(x) \leq 100r + \varepsilon$
- Constraint 2: $\frac{1}{|S|} \sum_{x \in S} \tilde{\psi}_{p,T,10r}(x) \leq \varepsilon$

To see that these constraints are indeed convex, first note that, by definition of $\tilde{\phi}_{p,T,10r}(x)$ and the triangle inequality, we have for any pair of polynomials $p_1, p_2$, $x$ in $R^d$ and $\alpha \in [0, 1]$

$$\tilde{\phi}_{(\alpha p_1 + (1-\alpha)p_2),T,10r}(x) \leq \alpha\tilde{\phi}_{p_1,T,10r}(x) + (1 - \alpha)\tilde{\phi}_{p_2,T,10r}(x), \tag{14}$$

which implies that Constraint 1 is convex. Likewise, reclalling the definition of $\psi$

$$\tilde{\psi}_{p,T,10r} := 10 \cdot \mathbb{1}[\tilde{\phi}(x)_{p,T,10r} > 0.6] \cdot (\tilde{\phi}(x)_{p,T,10r} - 0.6),$$

and combining it with Equation (14) we see that

$$\tilde{\psi}_{\alpha p_1 + (1-\alpha)p_2,T,10r}(x) \leq \alpha\tilde{\psi}_{p_1,T,10r}(x)(1 - \alpha)\tilde{\psi}_{p_2,T,10r}(x).$$

We now proceed to the proof of correctness, the following claim will be key to our argument, which will be proven in the next subsection (Appendix B.4):

**Claim B.11.** *For any specific iteration $i$ of the algorithm, the polynomial $p_i$ will satisfy the following:*

$$\mathbb{E}_{(x,y) \sim S_i} \left[ \tilde{\phi}_{p_i,T_i,10r}(x) \right] \leq 100r + O(\varepsilon), \tag{15}$$

$$\mathbb{E}_{(x,y) \sim S_i} \left[ \tilde{\psi}_{p_i,T_i,10r}(x) \right] \leq O(\varepsilon), \tag{16}$$

*and the following holds with probability at least $0.9$:*

$$\mathbb{E}_{(x,y) \sim S_i} \left[ \frac{|p_i(x) - y|}{2} \right] \leq \min_{f \in \mathcal{F}} \left( \Pr_{(x,y) \sim D} [f(x) \neq y] \right) + O(\varepsilon), \tag{17}$$

We now prove Theorem 3.3 assuming Claim B.11.

*Proof of Theorem 3.3.* First we note that for sufficiently large value of the absolute constant $C$, with probability at least $1 - O(\delta)$ for all $i$ in $\{1, \cdots, \log 1/\delta\}$, the set $S_i$ will satisfy Claim E.2, Claim E.3 and Fact E.5, and the set $T_i$ will satisfy Claim E.2. Furthermore, since the main iteration is repeated

$O(\log 1/\delta)$ times, the we see that with probability $1 - O(\delta)$ the polynomial $p = p_{i^*}$ will satisfy Equations 15, 16 and 17. Denoting $S = S_{i^*}$ and $T = T_{i^*}$, we have

$$\mathbb{E}_{t\sim[-1,1]}\left[\mathbb{E}_{(x,y)\sim D}\left[\phi_{\text{sign}(p-t),10r}(x)\right]\right] \overbrace{\leq \mathbb{E}_{t\sim[-1,1]}\left[\mathbb{E}_{(x,y)\sim S}\left[\phi_{\text{sign}(p-t),10r}(x)\right]\right] + O(\varepsilon)}^{\text{Since } S \text{ satisfies Claim E.2}} \leq$$

$$\leq \underbrace{\mathbb{E}_{t\sim[-1,1]}\left[\mathbb{E}_{(x,y)\sim S}\left[\tilde{\phi}_{\text{sign}(p-t),T,10r}(x)\right]\right] + O(\varepsilon)}_{\text{Since } T \text{ satisfies Claim A.3, and WLOG } \varepsilon < 0.02.} \leq \underbrace{\mathbb{E}_{(x,y)\sim S}\left[\tilde{\phi}_{p,T,10r}(x)\right] + O(\varepsilon)}_{\text{By Fact A.4}} \underbrace{\leq 100r + O(\varepsilon)}_{\text{By Equation (15)}},$$

which gives us Equation (2). We now derive Equation (3) as follows:

$$\Pr_{(x,y)\sim\mathcal{D}}\left[\mathbb{E}_{t\in[-1,1]}[\phi_{\text{sign}(p(x)-t),10r}] \geq 0.7\right] \overbrace{\leq \Pr_{(x,y)\sim S}[\mathbb{E}_{t\in[-1,1]}[\phi_{\text{sign}(p-t),10r}(x)] \geq 0.67] + O(\varepsilon)}^{\text{Since } S \text{ satisfies Claim E.3}} \leq$$

$$\leq \underbrace{\Pr_{(x,y)\sim S}[\mathbb{E}_{t\sim[-1,1]}[\tilde{\phi}_{\text{sign}(p-t),T,10r}(x)] \geq 0.65] + O(\varepsilon)}_{\text{Since } T \text{ satisfies Claim A.3, and WLOG } \varepsilon < 0.02.} \leq \underbrace{O(1)\,\mathbb{E}_{(x,y)\sim S}\left[\tilde{\psi}_{p,T,10r}(x)\right] + O(\varepsilon).}_{\text{By Fact A.4 and definition of } \tilde{\psi} \text{ (Section 2.1)}} \underbrace{\leq O(\varepsilon)}_{\text{By Equation (16)}}$$

Finally, we derive Equation (4) as follows:

$$\mathbb{E}_{t\sim[-1,1]}\left[\text{err}_D\left[\text{sign}(p-t)\right]\right] \overbrace{\leq \mathbb{E}_{t\sim[-1,1]}\left[\widehat{\text{err}}_S\left[\text{sign}(p-t)\right]\right] + O(\varepsilon)}^{\text{Since } S \text{ satisfies Fact E.5}} \leq$$

$$\leq \underbrace{\mathbb{E}_{(x,y)\sim S}\left[\frac{|p(x)-y|}{2}\right] + O(\varepsilon)}_{\text{By Fact A.4}} \underbrace{\leq \text{opt} + O(\varepsilon)}_{\text{By Equation (17) and defn of opt}} .$$

$\square$

## B.4 Proving Claim B.11

Finally, we finish this section by proving Claim B.11. Inspecting the algorithm LearnRealValued, we see that the linear program is always feasible for any input dataset $S_i$, because the all-zeros polynomial will satisfy the constraints in the linear program (i.e. Equations 15 and 16). It remains to show that with probability at least $0.9$ there exists a polynomial $p_i$ satisfying all three of Equations 15, 16 and 17.

Suppose $f^*$ is the optimal halfspace for $D$, i.e.

$$\Pr_{(x,y)\sim D}[f^*(x) \neq y] = \min_{f\in\mathcal{F}}\left(\Pr_{(x,y)\sim D}[f(x) \neq y]\right). \tag{18}$$

Let $D_{\text{marginal}}$ denote the $\mathbb{R}^d$-marginal of $D$, and let $D'_{\text{marginal}}$ denote distribution of $x + 10rz$, where $x$ is sampled from $D_{\text{marginal}}$ and $z$ is sampled from $\mathcal{N}(0, I_d)$. Since $D_{\text{marginal}}$ is assumed to be sub-Gaussian log-concave, Fact B.1 and Fact B.2 tell us that $D'_{\text{marginal}}$ is also sub-Gaussian log-concave. Furthermore, since $D_{\text{marginal}}$ is isotropic (i.e. has mean zero and identity convariance) and $r$ is in $(0,1)$ we see that

$$I_d \preceq \mathbb{E}_{x\sim D'_{\text{marginal}}}[xx^T] \preceq 101 I_d.$$

Therefore, Claim B.8 tells us that for a sufficiently large absolute constant $C'$ there is a polynomial of $p^*$ of degree $\frac{C'\log^2(1/\varepsilon)}{\varepsilon^2}$ for which

$$\mathbb{E}_{x\sim D_{\text{marginal}}}\left[|f^*(x) - p^*(x)|\right] \leq O(\varepsilon) \tag{19}$$

$$\mathbb{E}_{x\sim D'_{\text{marginal}}}\left[|f^*(x) - p^*(x)|\right] \leq O(\varepsilon) \tag{20}$$

Recall that for each element $(x,y)$ in $S_i$, the variable $x$ is distributed according to $D_{\text{marginal}}$. Similarly, for each $(x,y)$ in $S_i$ and $z$ in $T_i$ the sum $x + 10rz$ is distributed $D'_{\text{marginal}}$. This allows us to use inequalities above to conclude that with probability at least $0.99$ we have

$$\mathbb{E}_{(x,y)\sim S_i}\left[|p^*(x) - f^*(x)|\right] \leq O(\varepsilon) \tag{21}$$

$$\mathbb{E}_{(x,y)\sim S_i, z\sim T_i}\left[|p^*(x + 10rz) - f^*(x + 10rz)|\right] \leq O(\varepsilon) \tag{22}$$

In the remainder of this section we show that with probability at least $0.9$ over the choice of $S_i$ and $T_i$ the choice $p_i = p^*$ will indeed satisfy Equations 15, 16 and 17.

**B.4.1 Bounding $\mathbb{E}_{x \sim D_{\mathbf{marginal}}}[\phi_{f^*,10r}(x)]$ and $\mathbb{E}_{x \sim D_{\mathbf{marginal}}}[\psi_{f^*}(x)]$**

Recall that $f^*(x) = \text{sign}(u \cdot x - \tau)$ for some unit vector $u$ and real $\tau$. Here, we show that $f^*$ satisfies the following two inequalities

$$\mathbb{E}_{x \sim D_{\mathrm{marginal}}}[\phi_{f^*,10r}(x)] \leq 60r, \tag{23}$$

$$\mathbb{E}_{x \sim D_{\mathrm{marginal}}}[\psi_{f^*}(x)] = 0. \tag{24}$$

Indeed, fix a specific value of $x$. Recall that $\phi_{f^*,10r}(x)$ is the probability that $\text{sign}(u \cdot x - \tau) \neq \text{sign}(u \cdot (x + 10rz) - \tau)$ where $z$ is sampled from $\mathcal{N}(0, I_d)$. With probability at least $0.5$ over the choice of $Z$ we have $\text{sign}(u \cdot z) = \text{sign}(u \cdot x - \tau)$ which leands to $\text{sign}(u \cdot x - \tau) \neq \text{sign}(u \cdot (x + 10rz) - \tau)$. Thus, $\phi_{f^*,10r}(x) \leq 0.5$ and substituting this into the definition of $\psi$ we see that $\psi(x) = 0$, which proves Equation (24).

Now consider points $x$ in $\mathbb{R}^d$ such that $|u \cdot x - \tau|$ is in the interval $[10r \cdot (i-1), 10r \cdot i]$ for an integer $i$. We have the following two observations:

- By Fact B.4, we see that the probability over $x$ sampled from $D_{\mathrm{marginal}}$ that $|u \cdot x - \tau|$ is in the interval $[10r \cdot (i-1), 10r \cdot i]$ can be upper-bounded by $20r$.

- For a fixed $x$ for which $|u \cdot x - \tau|$ is in the interval $[10r \cdot (i-1), 10r \cdot i]$, via Chebyshev's inequality, it follows that

$$\Pr_{z \sim \mathcal{N}(0, I_d)}[\text{sign}(u \cdot x - \tau) \neq \text{sign}(u \cdot (x + 10rz) - \tau)] \leq \max\left(1, \left(\frac{10r}{10r(i-1)}\right)^2\right) = \max\left(1, \left(\frac{1}{i-1}\right)^2\right)$$

Combining the two observations above, we see that

$$\mathbb{E}_{x \sim D_{\mathrm{marginal}}}[\phi_{f^*,10r}(x)] \leq \sum_{i=0}^{\infty} 20r \max\left(1, \left(\frac{1}{i-1}\right)^2\right) \leq 60r,$$

which yields Equation (23).

**B.4.2 Bounding $\mathbb{E}_{(x,y) \sim S_i}[\tilde{\phi}_{p^*,T_i,10r}(x)]$ and $\mathbb{E}_{(x,y) \sim S_i}[\tilde{\psi}_{p^*,T_i,10r}(x)]$.**

We observe that

$$\mathbb{E}_{(x,y) \sim S_i}[\tilde{\phi}_{p^*,T_i}(x)] = \mathbb{E}_{\substack{(x,y) \sim S_i \\ z \sim T_i}}\left[\left|\frac{p^*(x) - p^*(x + 10rz)}{2}\right|\right] \leq$$

$$\mathbb{E}_{\substack{(x,y) \sim S_i \\ z \sim T_i}}\left[\left|\frac{f^*(x) - f^*(x + 10rz)}{2}\right|\right] + \mathbb{E}_{(x,y) \sim S_i}\left[\left|\frac{p^*(x) - f^*(x)}{2}\right|\right] + \mathbb{E}_{\substack{(x,y) \sim S_i \\ z \sim T_i}}\left[\left|\frac{p^*(x + 10rz) - f^*(x + 10rz)}{2}\right|\right] \leq$$

$$\mathbb{E}_{(x,y) \sim S_i}[\tilde{\phi}_{f^*,T_i,10r}(x)] + \underbrace{\mathbb{E}_{(x,y) \sim S_i}|p^*(x) - f^*(x)| + \mathbb{E}_{(x,y) \sim S_i, z \sim T_i}|p^*(x + 10rz) - f^*(x + 10rz)|}_{=O(\varepsilon) \text{ by Equations 21 and 22}}.$$

$$\tag{25}$$

Analogously, by inspecting the definition of $\psi$, we see that for any $x$ we have

$$\left|\tilde{\psi}_{p^*,T_i}(x) - \tilde{\psi}_{f^*,T_i}(x)\right| \leq |f^*(x) - p^*(x)| + O(1) \cdot \mathbb{E}_{z \sim T_i}\left[|f^*(x + 10z) - p^*(x + 10z)|\right]$$

Averaging over $x$ in $S_i$, we have

$$\mathbb{E}_{(x,y) \sim S_i}[\tilde{\psi}_{p^*,T_i}(x)] \leq$$

$$\mathbb{E}_{(x,y) \sim S_i}[\tilde{\phi}_{f^*,T_i,10r}(x)] + O(1) \cdot \underbrace{\left(\mathbb{E}_{(x,y) \sim S_i}|p^*(x) - f^*(x)| + \mathbb{E}_{(x,y) \sim S_i, z \sim T_i}|p^*(x + 10rz) - f^*(x + 10rz)|\right)}_{=O(\varepsilon) \text{ by Equations 21 and 22}}.$$

$$\tag{26}$$

Recall that $f^*$ is a $\{\pm 1\}$-valued function, which implies that $\tilde{\phi}_{f^*,T_i,10r}(x + 10rz)$ is in $[0, 1]$ for all $x$ and $z$. This allows us to use the Chebyshev's inequality to conclude that:

$$\mathbb{E}_{T_i}\left[\left|\tilde{\phi}_{f^*,T_i,10r}(x) - \phi_{f^*,10r}(x)\right|\right] \leq O\left(\frac{1}{\sqrt{|T_i|}}\right).$$

Inspecting the definition of $\tilde{\psi}$, we see that for any $x$ it is the case that

$$\left|\tilde{\psi}_{f^*,T_i}(x) - \psi_{f^*}(x)\right| \leq O(1) \cdot \left|\tilde{\phi}_{f^*,T_i,10r}(x) - \phi_{f^*,10r}(x),\right|$$

which implies that

$$\mathbb{E}_{T_i}\left[\left|\tilde{\psi}_{f^*,T_i}(x) - \psi_{f^*}(x)\right|\right] \leq O\left(\frac{1}{\sqrt{|T_i|}}\right)$$

Averaging the inequalities above over $x$ in $S_i$, we have:

$$\mathbb{E}_{T_i}\left[\mathbb{E}_{(x,y)\sim S_i}\left[\left|\tilde{\phi}_{f^*,T_i,10r}(x) - \phi_{f^*,10r}(x)\right|\right]\right] \leq O\left(\frac{1}{\sqrt{|T_i|}}\right)$$

$$\mathbb{E}_{T_i}\left[\mathbb{E}_{(x,y)\sim S_i}\left[\left|\tilde{\psi}_{f^*,T_i}(x) - \psi_{f^*}(x)\right|\right]\right] \leq O\left(\frac{1}{\sqrt{|T_i|}}\right)$$

Thus, with probability at least $0.99$ over the choice of $T_i$, we have

$$\left|\mathbb{E}_{(x,y)\sim S_i}\left[\tilde{\phi}_{f^*,T_i,10r}(x)\right] - \mathbb{E}_{(x,y)\sim S_i}\left[\phi_{f^*,10r}(x)\right]\right| \leq \mathbb{E}_{(x,y)\sim S_i}\left[\left|\tilde{\phi}_{f^*,T_i,10r}(x) - \phi_{f^*,10r}(x)\right|\right] \leq O\left(\frac{1}{\sqrt{|T_i|}}\right)$$
$$\tag{27}$$

$$\left|\mathbb{E}_{(x,y)\sim S_i}\left[\tilde{\psi}_{f^*,T_i}(x)\right] - \mathbb{E}_{(x,y)\sim S_i}\left[\psi_{f^*}(x)\right]\right| \leq \mathbb{E}_{(x,y)\sim S_i}\left[\left|\tilde{\psi}_{f^*,T_i}(x) - \psi_{f^*}(x)\right|\right] \leq O\left(\frac{1}{\sqrt{|T_i|}}\right)$$
$$\tag{28}$$

Again, recalling that $f^*$ is a $\{\pm1\}$-valued function, we see that $\phi_{f^*,10r}(x)$ is in $[0,1]$ for all $x$ and $\psi_{f^*}(x)$ is likewise bounded by $O(1)$ in absolute value. This allows us to use the Chebyshev's inequality to conclude that with probability at least $0.99$ over the choice of $S_i$ we have:

$$\left|\mathbb{E}_{(x,y)\sim S_i}\left[\phi_{f^*,10r}(x)\right] - \mathbb{E}_{x\sim D_{\text{marginal}}}\left[\phi_{f^*,10r}(x)\right]\right| \leq O\left(\frac{1}{\sqrt{|S_i|}}\right) \tag{29}$$

$$\left|\mathbb{E}_{(x,y)\sim S_i}\left[\psi_{f^*}(x)\right] - \mathbb{E}_{x\sim D_{\text{marginal}}}\left[\psi_{f^*}(x)\right]\right| \leq O\left(\frac{1}{\sqrt{|S_i|}}\right) \tag{30}$$

Overall, combining Equation (23), Equation (29), Equation (27) and Equation (25) we see that with probability at least $0.97$ over the choice of $S_i$ and $T_i$ it is the case that

$$\mathbb{E}_{(x,y)\sim S_i}[\tilde{\phi}_{p^*,10r}(x)] \leq 60r + O(\varepsilon).$$

Similarly, combining Equation (24), Equation (30), Equation (28) and Equation (26) we see that with probability at least $0.97$ over the choice of $S_i$ and $T_i$ it is the case that

$$\mathbb{E}_{(x,y)\sim S_i}[\tilde{\psi}_{p^*,T_i}(x)] \leq O(\varepsilon).$$

### B.4.3 $p^*$ has a small empirical error.

Finally, we argue that with probability $0.99$ we have $\mathbb{E}_{(x,y)\sim S_i}\left[\frac{|p^*(x)-y|}{2}\right] \leq \mathbb{E}_{(x,y)\sim D}\left[\frac{|f^*(x)-y|}{2}\right] + O(\varepsilon)$. Indeed, Hoeffding's inequality implies that with probability at least $0.995$ we have

$$\mathbb{E}_{(x,y)\sim S_i}\left[\frac{|f^*(x)-y|}{2}\right] \leq \mathbb{E}_{(x,y)\sim D}\left[\frac{|f^*(x)-y|}{2}\right] + \varepsilon,$$

whereas Markov's inequality tells us that with probability $0.995$ we have

$$\mathbb{E}_{(x,y)\sim S_i}\left[\frac{|f^*(x)-p^*(x)|}{2}\right] \leq O(1) \cdot \mathbb{E}_{(x,y)\sim D}\left[\frac{|f^*(x)-p^*(x)|}{2}\right] = O(\varepsilon).$$

Overall, we see that with probability at least $0.99$ it holds that

$$E_{(x,y)\sim S_i}\left[\frac{|p^*(x)-y|}{2}\right] \leq E_{(x,y)\sim S_i}\left[\frac{|f^*(x)-y|}{2}\right] + E_{(x,y)\sim S_i}\left[\frac{|p^*(x)-f^*(x)|}{2}\right] \leq \mathbb{E}_{(x,y)\sim D}\left[\frac{|f^*(x)-y|}{2}\right] + O(\varepsilon)$$

# C  Pseudocode and analysis of ComputeClassifier

## C.1  Local correction of adversarial robustness

**Theorem C.1** (Correctness and complexity of Algorithm 3). *Let $\hat{\phi}$ be an $\varepsilon$-accurate local noise sensitivity approximator for degree-$k$ PTFs over $\mathbb{R}^d$, and let $\tau$ be the run-time and query complexity of $\hat{\phi}$. Then there exists an algorithm ROBUSTNESSLCA$(x, g, r)$ that takes $x$ in $\mathbb{R}^d$, query access to a function $g : \mathbb{R}^d \to \{-1, 1\}$, perturbation size parameter $r$ in $(0, 1)$. The algorithm makes $O(\tau)$ queries to $g$, runs in time $O(\tau + d)$, and satisfies the specifications below.*

*Let $\mathcal{D}$ be a distribution over $\mathbb{R}^d$, and let $g$ be a degree-$k$ PTF. Let the function $h : \mathbb{R}^d \to \{-1, 1\}$ be defined as $h(x) := \text{ROBUSTNESSLCA}(x, g, r)$. Then the following properties hold:*

*a) $\Pr_{x \sim \mathcal{D}}[g(x) \neq h(x)] \leq \widehat{\text{iso}}_{\mathcal{D}, 10r}(g, 0.8)$.*

*b) $\Pr_{x \sim \mathcal{D}}[\phi_{g, 10r}(x) \leq 0.1] \geq 1 - O\left(\widehat{\text{NS}}_{\mathcal{D}, 10r}(g)\right)$.*

*c) For every $x$ in $\mathbb{R}^d$, if $\hat{\phi}_{g, 10r}(x) \leq 0.1$, then for every $x'$ with $\|x' - x\| \leq r$ we have $h(x') = h(x)$.*

---

**Algorithm 3** ROBUSTNESSLCA$(x, g, r)$:

---

1: **Input:** point $x \in \mathbb{R}^d$, black-box representation of a function $g : \mathbb{R}^d \to \{-1, 1\}$,
2:     robustness radius $r$
3: **Uses:** local noise sensitivity approximator $\hat{\phi}$.                    (See Definition 2.5)
4: **Output:** $b \in \{-1, 1\}$
5: **if** $\hat{\phi}_{g, 10r}(x) > 0.8$ **then**
6:     **return** $-g(x)$
7: **else**
8:     **return** $g(x)$

---

*Proof.* Inspecting the algorithm ROBUSTNESSLCA, we see that condition (a) follows directly, as only isolated points have their labels changed. Meanwhile, condition (b) follows directly from Markov's inequality and the definition of $\widehat{\text{NS}}$ (Section 2.1.1).

To show condition (c), we first recall that by the assumption that $\hat{\phi}$ is an $\varepsilon$-accurate approximator, for every degree-$k$ polynomial threshold function $g$ and every $x$ in $\mathbb{R}^d$ we have

$$\left|\hat{\phi}_{g, 10r}(x) - \phi_{g, 10r}(x)\right| \leq \varepsilon \leq 0.01, \tag{31}$$

where for the last we assumed without loss of generality that $\varepsilon < 0.01$. This implies that whenever $\hat{\phi}_{g, 10r}(x) \leq 0.1$ we have $\phi_{g, 10r}(x) \leq 0.11$. Suppose $x'$ has distance at most $r$ from $x$. Since $\phi_{g, r}(x)$ is defined as

$$\phi_{g, 10r}(x) := \Pr_{z \sim N(0, I_d)}[g(x) \neq g(x + 10rz)],$$

we see that

$$\Pr_{z \sim N(0, I_d)}[g(x) \neq g(x' + 10rz)] \leq$$

$$\Pr_{z \sim N(0, I_d)}[g(x) \neq g(x + 10rz)] + d_{TV}(N(x, 100r^2 I_d), N((x'), 100r^2 I_d)) \leq$$

$$\Pr_{z \sim N(0, I_d)}[g(x) \neq g(x + 10rz)] + \frac{\|x - x'\|}{20r},$$

where the last inequality is implied by Fact A.5. Overall, since we know that $\phi_{g, 10r}(x) \leq 0.11$ and $\|x - x'\| \leq r$, we have

$$\Pr_{z \sim N(0, I_d)}[g(x) \neq g(x' + 10rz)] \leq 0.16.$$

Now, we consider the following two cases:

- Suppose $g(x') = g(x)$: then the above implies that $\phi_{g,10r}(x') \leq 0.16$, which combined with Equation (31) implies that $\hat{\phi}_{g,10r}(x') \leq 0.17$. Therefore, by inspecting the algorithm ROBUSTNESSLCA we see that $h(x') = g(x') = g(x) = h(x)$.

- On the other hand, suppose $g(x') = -g(x)$. Then, we have $\phi_{g,10r}(x') \geq 1 - 0.16 = 0.84$, which combined with Equation (31) implies that $\hat{\phi}_{g,10r}(x') \geq 0.83$. Therefore, by inspecting the algorithm ROBUSTNESSLCA we see that $h(x') = -g(x') = g(x) = h(x)$.

In either case, we have $h(x') = h(x)$.

$\square$

## C.2 Finding good rounding thresholds

**Theorem C.2** (Correctness of Algorithm 4). *Let $\varepsilon \in (0,1)$ and $\mathcal{D}$ be a distribution over $\mathbb{R}^d \times \{\pm 1\}$. Let $\hat{\phi}$ be an $\varepsilon$-accurate local noise sensitivity approximator for degree-$k$ PTFs over $\mathbb{R}^d$. Let $p : \mathbb{R}^d \to \mathbb{R}$ be a polynomial of degree $\leq k$ satisfying the following:*

$$\mathop{\mathbb{E}}_{t \sim [-1,1]}\left[\mathop{\mathbb{E}}_{(x,y) \sim \mathcal{D}}[\phi_{\mathrm{sign}(p-t),10r}(x)]\right] \leq 100r + O(\varepsilon), \tag{32}$$

$$\Pr_{(x,y) \sim \mathcal{D}}\left[\mathbb{E}_{t \sim [-1,1]}[\phi_{\mathrm{sign}(p(x)-t),10r}(x) \geq 0.7]\right] \leq O(\varepsilon). \tag{33}$$

*The algorithm* COMPUTEROUNDINGTHRESHOLDS *outputs real numbers* $t_1, t_2, t_3, t_4$ *and* $w_1, w_2, w_3, w_4 \in [0,1]$ *such that* $\sum_i w_i = 1$. *Let the hypotheses* $g_1, g_2, g_3, g_4$ *be defined as*

$$g_i(x) = \mathrm{sign}(p(x) - t_i).$$

*Then the following properties hold with probability at least* $1 - O(\delta)$:

- $\sum_i w_i \cdot \mathrm{err}_{\mathcal{D}}(g_i) \leq \mathbb{E}_{t \sim [-1,1]}[\mathrm{err}_{\mathcal{D}}(\mathrm{sign}(p-t))] + O(\varepsilon)$

- $\sum_i w_i \cdot \widehat{\mathrm{NS}}_{\mathcal{D},10r}(g_i) \leq 200r + O(\varepsilon)$.

- $\sum_i w_i \cdot \widehat{\mathrm{iso}}_{\mathcal{D},10r}(g_i, 0.8) \leq O(\varepsilon)$.

We argue that with high probability over the choice of the $O(1/\varepsilon^2)$ random rounding thresholds, the equal-weighted mixture of the rounded functions has each of the desired properties.

**Claim C.3** (Properties of the equal mixture of all rounding thresholds). *Let* $\varepsilon \in (0,1)$. *Let* $p : \mathbb{R}^d \to \mathbb{R}$ *be a polynomial of degree* $\leq k$ *satisfying the following:*

- $\mathbb{E}_{t \sim [-1,1]}\left[\mathbb{E}_{(x,y) \sim \mathcal{D}}[\hat{\phi}_{\mathrm{sign}(p-t),10r}(x)]\right] \leq \alpha$

- $\Pr_{(x,y) \sim \mathcal{D}}\left[\mathbb{E}_{t \in [-1,1]}[\hat{\phi}_{\mathrm{sign}(p(x)-t),10r}(x) \geq 0.75\right] \leq O(\varepsilon)$.

*Let* $\Theta$ *be a set of* $100/\varepsilon^2$ *real numbers drawn uniformly and independently from* $[-1,1]$. *Then the following hold with probability* $\geq 0.99$:

(a) $\mathbb{E}_{t \in \Theta}[\mathrm{err}_{\mathcal{D}}(\mathrm{sign}(p-t))] \leq \mathbb{E}_{t \sim [-1,1]}[\mathrm{err}_{\mathcal{D}}(\mathrm{sign}(p-t))] + O(\varepsilon)$

(b) $\mathbb{E}_{t \in \Theta}[\mathbb{E}_{x \sim \mathcal{D}}[\hat{\phi}_{\mathrm{sign}(p-t),10r}(x)]] \leq 2\alpha + O(\varepsilon)$

(c) $\mathbb{E}_{t \in \Theta}[\Pr_{x \sim \mathcal{D}}[\hat{\phi}_{\mathrm{sign}(p-t),10r}(x) > 0.8]] \leq O(\varepsilon)$.

*Proof of Claim C.3.* We will show that each of the conditions holds with high probability, then union bound.

---

**Algorithm 4** COMPUTEROUNDINGTHRESHOLDS$(D, p, r, \varepsilon, \delta)$:

---

1: **Input:** sample access to $D$, robustness radius $r$,
2:     error bound $\varepsilon$, confidence bound $\delta$
3: **Uses:** local noise sensitivity approximator $\hat{\phi}$.                    (See Definition 2.5)
4: **Output:** thresholds $\vec{t} = t_1, t_2, t_3, t_4 \in [-1, 1]$ and weights $\vec{w} = w_1, w_2, w_3, w_4 \in [0, 1]$
5: Initialize $Q := \varnothing$.
6: $M \leftarrow 100/\varepsilon^3 \cdot \log^2(1/\delta)$ i.i.d. samples $(x, y)$ from $\mathcal{D}$
7: **for** $i \in [\log(1/\delta)]$ **do**
8:     **for** $j \in [100/\varepsilon^2]$ **do**
9:         Draw rounding threshold $t$ u.a.r from $[-1, 1]$ and define rounded function $p_t(x) := \text{sign}(p(x) - t)$.
10:         Let $\text{err}_t := \text{err}_M(p_t)$.
11:         Let $\text{NS}_t := \frac{1}{|M|} \cdot \sum_{x \in M} \hat{\phi}_{p_t, 10r}(x)$.
12:         Let $\text{iso}_t := \frac{1}{|M|} \cdot \sum_{x \in M} \mathbb{1}[\hat{\phi}_{p_t, 10r}(x) \geq 0.8]$.
13:         Let $q_t := (\text{err}_t, \text{NS}_t, \text{iso}_t)$.
14:         Add $q_t$ to $Q$.
15:     $\widehat{\text{opt}} := \frac{1}{|Q|} \sum_{(\text{err}_t, \text{NS}_t, \text{iso}_t) \in Q} (\text{err}_t)$
16:     **for** tuple $q_1, q_2, q_3, q_4 \in Q^4$ **do**
17:         Solve the linear program with variables $w_1, w_2, w_3, w_4$ defined by the following constraints:

  - $\sum_{i=1}^4 w_i = 1$
  - $\sum_{i=1}^4 w_i q_{i,1} \leq \widehat{\text{opt}} + C\varepsilon$                    $C$ is a sufficiently large absolute constant.
  - $\sum_{i=1}^4 w_i q_{i,2} \leq 200r + C\varepsilon$
  - $\sum_{i=1}^4 w_i q_{i,3} \leq C\varepsilon$

18:         If a solution is found, **return** $\vec{t} = t_1, t_2, t_3, t_4$ and $\vec{w} = w_1, w_2, w_3, w_4$.
19: **return** $\bot$

---

(a) Let $X = \sum_{t \in \Theta} \text{err}_{\mathcal{D}}(\text{sign}(p - t))$. We have $\mathbb{E}_\Theta[X] = |\Theta| \cdot \mathbb{E}_{t \sim [-1, 1]}[\text{err}_{\mathcal{D}}(\text{sign}(p - t))]$ and we apply Hoeffding's inequality:

$$\Pr\left[\mathbb{E}_{t \sim \Theta}[\text{err}_{\mathcal{D}}(\text{sign}(p - t))] > \mathbb{E}_{t \sim [-1,1]}[\text{err}_{\mathcal{D}}(\text{sign}(p - t))] + \varepsilon\right]$$
$$= \Pr[(X - \mathbb{E}[X])/|\Theta| > \varepsilon] \leq \exp(-2\varepsilon^2 \cdot 100/\varepsilon^2) \leq 0.001.$$

(b) We have by assumption that

$$\mathbb{E}_{t \sim [-1,1]} \mathbb{E}_{x \sim \mathcal{D}}[\hat{\phi}_{\text{sign}(p - t)}(x)] \leq \alpha.$$

Let $X = \sum_{t \in \Theta} \mathbb{E}_{x \sim \mathcal{D}}[\hat{\phi}_{\text{sign}(p-t), 10r}(x)]$, which has expectation $\leq |\Theta| \cdot \alpha$. Then by Hoeffding's inequality, we have

$$\Pr[X/|\Theta| > \alpha + \varepsilon] \leq \Pr[(X - \mathbb{E}[X])/|\Theta| > \varepsilon] \leq \exp(-2\varepsilon^2 \cdot 100/\varepsilon^2) \leq 0.001.$$

(c) We have by assumption that $\Pr_{x \sim \mathcal{D}}[\mathbb{E}_{t \sim [-1,1]}[\hat{\phi}_{\text{sign}(p-t)}](x) \geq 0.75] \leq O(\varepsilon)$. We want to bound

$$\mathbb{E}_{t \sim \Theta} \Pr_{x \sim \mathcal{D}}[\hat{\phi}_{\text{sign}(p-t)}(x) > 0.75].$$

We can upper bound this by

$$O(\varepsilon) + \mathbb{E}_{t \sim \Theta} \Pr_{x \sim \mathcal{D}}[\hat{\phi}_{\text{sign}(p-t)}(x) - \mathbb{E}_{t' \sim [-1,1]}[\hat{\phi}_{\text{sign}(p-t)}(x)] > 0.05] =$$
$$O(\varepsilon) + \Pr_{x \sim \mathcal{D}} \Pr_{t \sim \Theta}[\hat{\phi}_{\text{sign}(p-t)}(x) - \mathbb{E}_{t' \sim [-1,1]}[\hat{\phi}_{\text{sign}(p-t')}(x)] > 0.05].$$

Let $X = \sum_{t \in \Theta} \hat{\phi}_{\mathrm{sign}(p-t)}(x)$; by Hoeffding's inequality we have

$$
\Pr_{\Theta}\left[\mathbb{E}_{t \sim \Theta}[\hat{\phi}_{\mathrm{sign}(p-t)}(x)] > \mathbb{E}_{t' \sim [-1,1]}[\hat{\phi}_{\mathrm{sign}(p-t')}(x)] + 0.05\right] = \Pr[(X - \mathbb{E}[X]/|\Theta|] > 0.05
$$
$$
\leq \exp(-2 \cdot 0.025 \cdot (100/\varepsilon^2))
$$
$$
\leq \varepsilon/1000.
$$

Thus, combining all three probabilities, we have:

$$
\Pr_{x \sim \mathcal{D}} \Pr_{\Theta} \Pr_{t \sim \Theta}[\hat{\phi}_{\mathrm{sign}(p-t),10r}(x) - \mathbb{E}_{t' \sim [-1,1]}[\hat{\phi}_{\mathrm{sign}(p-t'),10r}(x)] > 0.05] \leq \varepsilon/1000
$$

Then we apply a Markov bound:

$$
\Pr_{\Theta}\left[\Pr_{x \sim \mathcal{D}, t \sim \Theta}[\hat{\phi}_{\mathrm{sign}(p-t),10r}(x) - \mathbb{E}_{t' \sim [-1,1]}[\hat{\phi}_{\mathrm{sign}(p-t'),10r}(x)] > 0.05] > \varepsilon\right] \leq 0.001
$$

Union bounding over all the conditions, we have that all three conditions hold simultaneously with probability at least $0.99$ over the choice of $\Theta$.

$\square$

Now we will show the existence of a mixture using four rounding thresholds from $\Theta$, and show that COMPUTEROUNDINGTHRESHOLDS finds it. We will make use of the following well-known theorem of Carathéodory.

**Theorem C.4** (Carathéodory's theorem [Car07]). *Let $P \subset \mathbb{R}^k$ be a set of points and $\mathrm{Conv}(P)$ be its convex hull. For any $p \in \mathrm{Conv}(P)$, there exists a set $S \subseteq P$ of $k + 1$ points such that $p$ can be written as a convex combination of points in $S$.*

*Proof of Theorem C.2.* We first note that since $\hat{\phi}$ is an $\varepsilon$-accurate local noise sensitivity approximator for degree-$k$ PTFs, Equations 32 and 33 respectively imply that[9]

$$
\mathbb{E}_{t \sim [-1,1]}\left[\mathbb{E}_{(x,y) \sim \mathcal{D}}[\hat{\phi}_{\mathrm{sign}(p-t),10r}(x)]\right] \leq 100r + O(\varepsilon), \tag{34}
$$

$$
\Pr_{\substack{x \sim \mathcal{D} \\ t \sim [-1,1]}}[\hat{\phi}_{\mathrm{sign}(p-t),10r}(x) > 0.75] \leq O(\varepsilon). \tag{35}
$$

Thus, the assumptions of Claim C.3 hold with $\alpha = 100r$. For a threshold $t$, let the three-dimensional point $q_t := (\mathrm{err}_t, \mathrm{NS}_t, \mathrm{iso}_t)$ be defined as in COMPUTEROUNDINGTHRESHOLDS. First we argue that each of these estimates is $O(\varepsilon)$-accurate. Each estimate is the expectation of a random variable bounded in $[0, 1]$, so by Hoeffding's inequality we have

$$
\Pr[|\mathrm{err}_t - \mathrm{err}_{\mathcal{D}}(\mathrm{sign}(p - t))| \geq \varepsilon] \leq \exp(-2\varepsilon^2 \cdot |M|),
$$

and likewise for each $\mathrm{NS}_t$ and $\mathrm{iso}_t$. We set $M$ large enough that this probability is $\leq \delta/2 \cdot \exp(-100/\varepsilon)$, giving us more than enough room to union bound over the $\log(1/\delta) \cdot 300/\varepsilon^2$ estimates. Likewise, by the Hoeffding bound, we have with probability $1 - \delta$ that

$$
\left|\widehat{\mathrm{opt}} - \mathbb{E}_{t \sim [-1,1]}[\mathrm{err}_{\mathcal{D}}(\mathrm{sign}(p - t))]\right| \leq \varepsilon.
$$

Thus with probability at least $1 - \delta$, all estimates are $\varepsilon$-accurate. Then, by Claim C.3, we have that with large constant probability over the randomness of the set $\Theta$, there exists a convex combination $q^\star$ of the points $\{q_t : t \in \Theta\}$ that satisfies the linear constraints

- $q^\star[1] \leq \mathbb{E}_{t \sim [-1,1]}[\mathrm{err}_{\mathcal{D}}(\mathrm{sign}(p - t))] + O(\varepsilon) \leq \mathrm{opt} + C\varepsilon$

- $q^\star[2] \leq 200r + C\varepsilon$

- $q^\star[3] \leq C\varepsilon$,

---

[9]We assume $\varepsilon \leq 0.05$ without loss of generality.

for a sufficiently large constant $C$. That linear combination is the equal-weighted mixture $\mathbb{E}_{t\in\Theta}[q_t]$. By Theorem C.4, since the $q_t$'s are points in $\mathbb{R}^3$, it is possible to write $q^\star$ as a convex combination of four members of $\{q_t : t \in \Theta\}$. Thus, for some 4-tuple in $Q^4$, the linear program is feasible. COMPUTEROUNDINGTHRESHOLDS searches every possible tuple, guaranteeing that a solution is returned.

By repeating $\log(1/\delta)$ times with independent draws of $\Theta$, the large constant success probability is boosted to $1 - \delta$. □

## C.3 Randomized set partitioning

---

**Algorithm 5** COMPUTECLASSIFIER$(\mathcal{D}, p, r, \varepsilon, \delta)$:

---

1: **Input:** sample access to $\mathcal{D}$, polynomial $p$,
2:         robustness radius $r$, error bound $\varepsilon$, confidence parameter $\delta$
3: **Uses:** local noise sensitivity approximator $\hat{\phi}$.       (See Definition 2.5)
4: **Output:** Classifier $h : \mathbb{R}^d \to \{\pm 1\}$.
5: $(t_1, t_2, t_3, t_4, w_1, w_2, w_3, w_4) \leftarrow$ COMPUTEROUNDINGTHRESHOLDS$(\mathcal{D}, p, r, \varepsilon, \delta)$.
         (See Algorithm 4).
6: $S_{\text{test}} \leftarrow C\log^2(1/\delta)/\varepsilon^2$ samples from $\mathcal{D}$ (for sufficiently large constant $C$)
7: **for** $i \in [\log(1/\delta)]$ **do**
8:      $M \leftarrow 10d^3$ i.i.d. samples $(x, y)$ from $\mathcal{D}$
9:      For $i \in \{1, ..., 4\}$, define the following Boolean functions: let

$$h_i(x) = \text{ROBUSTNESSLCA}(x, \text{sign}(p - t_i), r) \qquad \text{(See Algorithm 3)}$$

$$\text{RobustIndicator}_i(x) := \mathbb{1}[\hat{\phi}_{\text{sign}(p-t_i), 10r}(x) \leq 0.1]$$

10:      For each $i \in \{1, ..., 4\}$, let

$$\hat{\mu}_i \leftarrow \mathbb{E}_{(x,y)\in M} \mathbb{1}[y \neq h_i(x)]$$

$$\hat{\mu}'_i \leftarrow \mathbb{E}_{(x,y)\in M} \text{RobustIndicator}_i(x).$$

11:      Obtain an orthonormal collection of vectors $\{u_1, \cdots, u_{d/2}\}$ orthogonal to $\text{span}(\mu_1, \cdots, \mu_4, \mu'_1, \cdots, \mu'_4)$.
12:      Generate a uniformly random unit vector $u^\star$ in $\text{span}(u_1, \cdots, u_{d/2})$.
13:      Compute a partition of the real line into intervals $J_1, \ldots J_4$ of Gaussian mass $w_1 \ldots, w_4$ respectively.
14:      Check if the following hold for each $i \in [4]$:

$$\mathbb{E}_{(x,y)\sim S_{\text{test}}}[\mathbb{1}[y \neq h_i(x)] \wedge \langle x, u^\star \rangle \in J_i] = \mathbb{E}_{(x,y)\sim S_{\text{test}}}[\mathbb{1}[y \neq h_i(x)] \cdot w_i \pm C'\varepsilon$$

(for sufficiently large constant $C'$).

$$\mathbb{E}_{(x,y)\sim S_{\text{test}}}[\text{RobustIndicator}_i(x) \wedge \langle x, u^\star \rangle \in J_i] = \mathbb{E}_{x\sim S_{\text{test}}}[\text{RobustIndicator}_i(x)] \cdot w_i \pm C'\varepsilon)$$

15:      If so, **return** the function $h$ defined

$$h(x) = \sum_{i=1}^{4} h_i(x) \cdot \mathbb{1}[\langle u^\star, x \rangle \in J_i]$$

16: **return** $\bot$

---

In this section, we will prove the correctness of COMPUTECLASSIFIER. As explained in the previous section, the output of COMPUTERANDOMTHRESHOLDS is a list of four weights and four thresholds for rounding. The goal is for a small collection of sets — the sets of points misclassified by each threshold and the sets of points for which ROBUSTNESSLCA guarantees robustness — to each be partitioned such that $w_i$ of their mass falls into the $i^{th}$ part for each $i$.

The partition is a set of intervals along a unit vector chosen uniformly from the subspace orthogonal to the estimated mean vectors of the sets we want to partition. The intervals are chosen to have mass $w_1, \ldots, w_4$ under the one-dimensional standard Gaussian. We will argue that with high probability over the choice of the random unit vector, all of our sets will be approximately normally distributed in their projections on this vector, and thus the intervals will contain the right proportion of their mass. The key fact underlying this argument comes from the following theorem of [DHV06]:

**Theorem C.5** (Concentration of random projections: special case of Theorem 11 of [DHV06]). *Let $D$ be a distribution over $\mathbb{R}^d$ with mean zero and covariance bounded by $\lambda_{\max}$ in operator norm. Assume also that $D$ is supported only on points $x$ satisfying $\|x\|_2 \geq \nu\sqrt{d}$. For a unit vector $v$, let $D_v$ denote the distribution of $\langle x, v \rangle, x \sim D$, and let $D_{\|\cdot\|}$ denote the distribution of $\|x\|_2 / \sqrt{d}, x \sim D$. With probability at least*

$$1 - \frac{\lambda_{\max} \ln(1/\varepsilon)}{\varepsilon^3 \nu^2} \cdot \exp\left(-\Omega\left(\frac{\varepsilon^4 \nu^2 d}{\lambda_{\max} \ln(1/\varepsilon)}\right)\right)$$

*over a uniformly random unit vector $v$, the following holds for all intervals $J$:*

$$\left| \Pr_{z \sim D_v}[z \in J] - \Pr_{\substack{\sigma \sim D_{\|\cdot\|} \\ z \sim \mathcal{N}(0, \sigma^2)}}[z \in J] \right| \leq \varepsilon.$$

We apply this theorem to the distributions induced by a set we aim to partition. This is our underlying distribution $\mathcal{D}$ conditioned on a (potentially randomized) indicator, such as the indicator of a point being misclassified by a hypothesis; we will represent that indicator as $r(x)$. After projecting on the subspace orthogonal to our mean estimates, these distributions will still not have mean exactly zero, so we incorporate an error term into our analysis to account for that. The resulting claim is the following:

**Claim C.6** (Masses of intervals under random projections). *Let $r(x)$ be a randomized Boolean function over $\mathbb{R}^d$, $P$ be a $d/2$-dimensional linear subspace in $\mathbb{R}^d$, and $J \subseteq \mathbb{R}$ be an interval. Let $\mathcal{P}$ be a $d/2 \times d$ matrix whose rows are an orthonormal basis for $P$. Assume the following hold for some $\beta_1, \beta_2 \in (0, 1)$ and $\beta_3 \in ((d/2)^{-1/2}, 1)$:*

- ***Bounded mean:*** $\|\mathcal{P} \mathbb{E}_{x \sim \mathcal{D}}[r(x)x]\|_2 \leq \beta_1$.

- ***Thin-shell support:*** *Every $x$ for which $r(x)$ can be nonzero satisfies $\frac{\|\mathcal{P}x\|_2}{\sqrt{d/2}} = 1 \pm \beta_2$.*

- ***Large support:*** $\mathbb{E}_{x \sim \mathcal{D}}[r(x)] \geq \beta_3$.

*For unit vector $v$, let $\mathcal{D}_v^r$ denote the distribution of $\langle x, v \rangle, x \sim \mathcal{D}$ conditioned on $r(x) = 1$. With probability at least*

$$1 - \frac{(1 + \beta_1/\beta_3) \ln(1/\varepsilon)}{\varepsilon^3 \beta_3 (1 - 2\beta_2 - 2\beta_1)} \cdot \exp\left(-\Omega\left(\frac{\varepsilon^4 \beta_3 (1 - 2\beta_2 - 2\beta_1)d}{\ln(1/\varepsilon)(1 + \beta_1/\beta_3)}\right)\right)$$

*over $v$ drawn uniformly from unit vectors in $P$, we have*

$$\left| \Pr_{z \sim \mathcal{D}_v^r}[z \in J] - \Pr_{z \sim \mathcal{N}(0,1)}[z \in J] \right| \leq O(\beta_1/\beta_3 + \beta_2 + \varepsilon).$$

*Proof.* Let $\mu := \mathbb{E}_{x \sim \mathcal{D}, r}[x \mid r(x) = 1]$. We will define $D_{\mu, \mathcal{P}}^r$ to be the distribution of $\mathcal{P}(x - \mu)$ and $\mathcal{D}_{\|\cdot\|}$ to be the distribution of $\|\mathcal{P}(x - \mu)\|_2 / \sqrt{d/2}$, where $x$ is drawn from $\mathcal{D}$ conditioned on $r(x) = 1$.

We aim to apply Theorem C.5 to the zero-mean distribution $\mathcal{D}_{\mu, \mathcal{P}}^r$ in the $d/2$ dimensional space $P$, and find the appropriate bounds for $\nu$ and $\lambda_{\max}$.

- **Bounding $\nu$:** By the thin-shell assumption, every $x$ for which $r(x)$ can be nonzero satisfies $\|\mathcal{P}x\|_2 \geq \sqrt{d/2}(1 - \beta_2)$. Thus for any such $x$, we have $\|\mathcal{P}(x - \mu)\|_2 \geq \sqrt{d/2}(1 - \beta_2) - $

$\|\mathcal{P}\mu\|_2$ by triangle inequality. Now we just need to bound $\|\mathcal{P}\mu\|_2$:

$$\begin{aligned}
\|\mathcal{P}\mu\|_2 &= \left\| \mathcal{P} \mathop{\mathbb{E}}_{x,r}[x \mid r(x) = 1] \right\|_2 \\
&= \left\| \mathcal{P} \int_{x \in \mathbb{R}^n} x \cdot \frac{\mathcal{D}(x) \Pr_r[r(x) = 1]}{\Pr_{x,r}[r(x) = 1]} dx \right\|_2 \\
&= \left\| \mathcal{P} \frac{\mathbb{E}_{x,r}[r(x)x]}{\mathbb{E}_{x,r}[r(x)]} \right\|_2 \\
&= \frac{\|\mathcal{P} \mathbb{E}_{x,r}[r(x)x]\|_2}{\mathbb{E}_{x,r}[r(x)]} \leq \frac{\beta_1}{\beta_3}
\end{aligned}$$

by the bounded-mean and large support assumptions. This gives us $\nu \geq 1 - \beta_2 - \frac{\beta_1}{\beta_3\sqrt{d/2}}$, and thus $\nu^2 \geq 1 - 2\beta_2 - 2\frac{\beta_1}{\beta_3\sqrt{d/2}} \geq 1 - 2\beta_2 - 2\beta_1$, by the assumption that $\beta_3 \geq 1/\sqrt{d/2}$.

- **Bounding $\lambda_{\max}$:** We want a spectral bound on $\mathbb{E}_{x \sim \mathcal{D}_{\mu,\mathcal{P}}^r}[xx^T]$. We have:

$$\begin{aligned}
\mathop{\mathbb{E}}_{x \sim \mathcal{D}_{\mu,\mathcal{P}}^r}[xx^T] &= \frac{\mathbb{E}_{x \sim \mathcal{D},r}[r(x)(\mathcal{P}(x - \mu))(\mathcal{P}(x - \mu))^T]}{\mathbb{E}_{x,r}[r(x)]} \\
&\leq \frac{1}{\beta_3} \mathcal{P} \mathop{\mathbb{E}}_{x \sim \mathcal{D}} \left[ \left( xx^T + \mu\mu^T - x\mu^T - \mu x^T \right) \right] \mathcal{P}^T
\end{aligned}$$

Since $\mathcal{D}$ has mean zero, $\mathcal{P}$ and $\mathbb{E}[xx^T]$ has operator norm 1, and $\mathcal{P}\mu\mu^T\mathcal{P}^T$ has operator norm $\leq \|\mathcal{P}\mu\|_2$, we have

$$\lambda_{\max} = \left\| \mathop{\mathbb{E}}_{x \sim \mathcal{D}_{\mu,\mathcal{P}}^r}[xx^T] \right\|_{op} \leq \frac{1 + \|\mathcal{P}\mu\|_2}{\beta_3} \leq \frac{1 + \beta_1/\beta_3}{\beta_3}.$$

Substituting these parameters into Theorem C.5 gives us the claimed failure probability bound. The condition that holds with high probability is that for all intervals $J$, we have

$$\left| \Pr_{z \sim (\mathcal{D}_{\mu,\mathcal{P}}^r)_v}[z \in J] - \Pr_{\substack{\sigma \sim \mathcal{D}_{\|\cdot\|} \\ z \sim \mathcal{N}(0,\sigma^2)}}[z \in J] \right| \leq \varepsilon.$$

By the thin-shell assumption, the bound on $\|\mathcal{P}\mu\|_2$, and the triangle inequality, $\mathcal{D}_{\|\cdot\|}$ is supported on $[1 - \beta_1/\beta_3 - \beta_2, 1 + \beta_1/\beta_3 + \beta_2]$. Then by the TV distance bound for Gaussians, we have for any $\sigma$ in the support of $\mathcal{D}_{\|\cdot\|}$:

$$d_{TV}(\mathcal{N}(0,1), \mathcal{N}(0,\sigma^2)) \leq \frac{3|1 - \sigma^2|}{2} \leq O(\beta_1/\beta_3 + \beta_2),$$

so we have

$$\left| \Pr_{z \sim (\mathcal{D}_{\mu,\mathcal{P}}^r)_v}[z \in J] - \Pr_{z \sim \mathcal{N}(0,1)}[z \in J] \right| \leq O(\varepsilon + \beta_1/\beta_3 + \beta_2).$$

We now substitute the definition of $(\mathcal{D}_{\mu,\mathcal{P}}^r)_v$ and use the fact that $v$ is in $P$.

$$\begin{aligned}
\Pr_{z \sim (\mathcal{D}_{\mu,\mathcal{P}}^r)_v}[z \in J] &= \Pr_{x \sim \mathcal{D}^r}[\langle \mathcal{P}(x - \mu), v \rangle \in J] \\
&= \Pr_{x \sim \mathcal{D}^r}[\langle x, v \rangle + \langle \mu, v \rangle \in J] \\
&= \Pr_{y \sim \mathcal{D}_v^r}[y + \langle \mu, v \rangle \in J].
\end{aligned}$$

So we have

$$\left| \Pr_{z \sim \mathcal{D}_v^r}[z + \langle \mu, v \rangle \in J] - \Pr_{z \sim \mathcal{N}(0,1)}[z \in J] \right| \leq O(\varepsilon + \beta_1/\beta_3 + \beta_2).$$

By applying the uniform convergence property of Theorem C.5 for all intervals to shift $J$, we have

$$\left| \Pr_{z \sim \mathcal{D}_v^r}[z \in J] - \Pr_{z \sim \mathcal{N}(0,1)}[z - \langle \mu, v \rangle \in J] \right| \leq O(\varepsilon + \beta_1/\beta_3 + \beta_2).$$

By anticoncentration of the Gaussian distribution and the fact that $|\langle \mu, v \rangle| \leq \beta_1/\beta_3$, we have

$$\Pr_{z \sim \mathcal{N}(0,1)}[z - \langle \mu, v \rangle \in J] = \Pr_{z \sim \mathcal{N}(0,1)}[z \in J] \pm 2\beta_1/\beta_3.$$

Combining everything, we have

$$\left| \Pr_{z \sim \mathcal{D}_v^r}[z \in J] - \Pr_{z \sim \mathcal{N}(0,1)}[z \in J] \right| \leq O(\varepsilon + \beta_1/\beta_3 + \beta_2)$$

with probability at least

$$1 - \frac{(1 + \beta_1/\beta_3)\ln(1/\varepsilon)}{\varepsilon^3 \beta_3(1 - 2\beta_2 - 2\beta_1)} \cdot \exp\left( -\Omega\left( \frac{\varepsilon^4 \beta_3 (1 - 2\beta_2 - 2\beta_1)d}{\ln(1/\varepsilon)(1 + \beta_1/\beta_3)} \right) \right)$$

as desired. $\qquad\square$

We now argue that for $r$ indicating a set of large Gaussian volume and $\mathcal{P}$ being orthogonal to a good approximation to the mean of $r$, the assumptions of Claim C.6 are satisfied with small values of $\beta$.

The following facts are relevant to this proof.

**Fact C.7** (Thin-shell concentration of log-concave variables [GM11]). *Let $X$ be an isotropic random vector with log-concave density in $\mathbb{R}^d$. Then there are universal constants $c, C > 0$ such that for all $t \geq 0$,*

$$\Pr\left[ \left| \|X\|_2 - \sqrt{d} \right| \geq t \right] \leq C \exp(-cd^{1/2}\min((t/\sqrt{d})^3, t/\sqrt{d})).$$

**Corollary C.8.** *Let $X$ be an isotropic random vector with log-concave density in $\mathbb{R}^d$. Then there is a universal constant $C$ such that for any $\varepsilon > 0$,*

$$\Pr\left[ \left| \|X\|_2 - \sqrt{d} \right| \geq C(d\log(1/\varepsilon))^{1/3} \right] \leq \varepsilon.$$

**Fact C.9** (Thin-shell concentration of Gaussian variables (standard fact)). *Let $X$ be a standard Gaussian in $\mathbb{R}^d$. Then*

$$\Pr\left[ \left| \|X\|_2 - \sqrt{d} \right| \geq t \right] \leq 2\exp(-t/4).$$

**Claim C.10** (Log-concave distributions satisfy assumptions of Claim C.6). *Let $\mathcal{D}$ be an isotropic log-concave distribution over $\mathbb{R}^d$ and $K$ be a sufficiently large constant depending on those given by Fact C.7. Let $r$ be a randomized Boolean function such that $\mathbb{E}_{x \sim \mathcal{D}}[r(x)] \geq \beta$, and let $\hat{\mu}(r)$ be such that for all $i \in [d]$,*

$$\left| \hat{\mu}(r)_i - \left( \mathbb{E}_{x \sim \mathcal{D}}[r(x)x] \right)_i \right| \leq 1/d.$$

*Let $P$ be a $d/2$-dimensional linear subspace orthogonal to $\hat{\mu}(r)$, and $\mathcal{P}$ be a $d/2 \times d$ matrix whose rows are an orthonormal basis for $P$. Let $r_{\text{trunc}}$ be the truncation of $r$ defined as*

$$r_{\text{trunc}}(x) := r(x) \wedge \mathbb{1}\left[ \frac{\|\mathcal{P}x\|_2}{\sqrt{d/2}} = 1 \pm \frac{10Kd^{1/3}\ln d}{\sqrt{d/2}} \right].$$

*Then the following conditions hold:*

- **Bounded mean:** $\|\mathcal{P}\, \mathbb{E}_{x \sim \mathcal{D}}[r_{\text{trunc}}(x)x]\|_2 \leq O(\sqrt{1/d})$.

- **Thin-shell support:** *Every $x$ for which $r_{\text{trunc}}(x)$ can be nonzero satisfies $\frac{\|\mathcal{P}x\|_2}{\sqrt{d/2}} = 1 \pm \frac{10Kd^{1/3}\ln d}{\sqrt{d/2}}$.*

- **Large support:** $\mathbb{E}_{x \sim \mathcal{D}}[r_{\text{trunc}}(x)] \geq \beta - O(1/d)$.

*Furthermore,* $\Pr_{x\sim\mathcal{D}}[r(x) \neq r_{\text{trunc}}(x)] \leq O(1/d)$.

*Proof.* First we will prove the final inequality,

$$\Pr_{x\sim\mathcal{D}}[r(x) \neq r_{\text{trunc}}(x)] \leq O(1/d).$$

Since the projection of $\mathcal{D}$ by $\mathcal{P}$ is a log-concave distribution over $\mathbb{R}^{d/2}$ (Fact B.3), this inequality follows from from Corollary C.8 applied to $\mathcal{P}(\mathcal{D})$ with $\varepsilon = 1/d$. We now prove the other conditions.

- **Bounded mean:** Let

$$\mu(r_{\text{trunc}}) \coloneqq \mathop{\mathbb{E}}_{x\sim\mathcal{D}}[r_{\text{trunc}}(x)x] \quad \text{and} \quad \mu(r) \coloneqq \mathop{\mathbb{E}}_{x\sim\mathcal{D}}[r(x)x].$$

First we claim that for every $i$, $|\mu(r_{\text{trunc}})_i - \mu(r)_i| \leq 1/d$. We have

$$
\begin{aligned}
\mu(r)_i &= \mu(r_{\text{trunc}})_i + \mathbb{E}[r(x)x_i \cdot \mathbb{1}[r(x) \neq r_{\text{trunc}}(x)]] \\
&\leq \mu(r_{\text{trunc}})_i + \mathbb{E}[\|x\| \cdot \mathbb{1}[r(x) \neq r_{\text{trunc}}(x)] \\
&\leq \mu(r_{\text{trunc}})_i + \int_{t=10Kd^{1/3}\ln d}^{\infty} t \cdot C \exp(-ct^3/d)dt \\
&\leq \mu(r_{\text{trunc}})_i + C\int_{t=10Kd^{1/3}\ln d}^{\infty} t^3 \cdot \exp(-ct^3/d)dt \\
&\leq \mu(r_{\text{trunc}})_i + C\int_{u=(10K\ln d)^3 d}^{\infty} u \cdot \exp(-cu/d)du \\
&\leq \mu(r_{\text{trunc}})_i + \exp(-(10K\ln d)^3 \cdot c) \cdot ((10K\ln d)^3 \cdot c + 1) \cdot (d/c)^2 \\
&\leq \mu(r_{\text{trunc}})_i + 1/d. \qquad \text{(by setting } K \text{ sufficiently large)}
\end{aligned}
$$

A symmetric argument lower bounds $\mu(r)_i$ by $\mu(r_{\text{trunc}})_i - 1/d$. Now we bound $\|\mathcal{P}\mu(r_{\text{trunc}})\|_2$. We have $\|\mathcal{P}\mu(r_{\text{trunc}})\|_2 \leq \sup_{u\in P, \|u\|_2=1}\langle u, \mu(r_{\text{trunc}})\rangle$. Consider a unit vector $u \in P$. We have

$$
\begin{aligned}
\langle u, \mu(r_{\text{trunc}})\rangle &= \langle u, \mu(r_{\text{trunc}}) - \mu(r)\rangle + \langle u, \mu(r) - \hat\mu(r)\rangle + \langle u, \hat\mu(r)\rangle \\
&\leq \sum_{i\in[d]}\left|\frac{u_i}{d}\right| + \sum_{i\in[d]}\left|\frac{u_i}{d}\right| + 0 \\
&\leq 2/\sqrt{d}.
\end{aligned}
$$

- **Thin-shell support:** This follows immediately from the definition of $r_{\text{trunc}}$.

- **Large support:** This follows immediately from the assumption that $\mathbb{E}_{x\sim\mathcal{N}(0,I_d)}[r(x)] \geq \beta$ and the fact that $\Pr_{x\sim\mathcal{N}(0,I_d)}[r(x) \neq r_{\text{trunc}}(x)] \leq O(1/d)$.

$\qquad\qquad\qquad\qquad\qquad\qquad\qquad\qquad\qquad\qquad\qquad\qquad\qquad\qquad\qquad\qquad\qquad\qquad\qquad\square$

**Claim C.11** (Accuracy of the mean estimates). *Let $S$ be a set of $10d^3$ points drawn from a distribution $\mathcal{D}$ with covariance $I_d$. Let $\mathcal{F}$ be a collection of eight (possibly randomized) Boolean functions over $\mathbb{R}^d$. Then with probability at least $1 - O(d^{-3})$, the following holds for all $f \in \mathcal{F}$ and $i \in [d]$:*

$$\left|\frac{1}{T}\sum_{x\in T}x_if(x) - \mathop{\mathbb{E}}_{x\sim\mathcal{D}}x_if(x)\right| \leq 1/d.$$

*Proof.* By the fact that $\mathcal{D}$ has covariance $I_d$, we have $\text{Var}(x_i) = 1$ and $\text{Var}(\frac{1}{|T|}\sum x_i) = 1/|T|$, so by Chebyshev's inequality

$$\Pr_T\left[\left|\mathop{\mathbb{E}}_{x\sim\mathcal{D}}[x_if(x)] - \frac{1}{|T|}\sum_{x\in T}x_if(x)\right| > \frac{10d^2}{|T|}\right] \leq 0.01d^{-4}.$$

Set $|T| := 10d^3$; then the deviation becomes $\frac{10d^2}{|T|} = \frac{1}{d}$. Union bounding over $\mathcal{F}$ and $i \in [d]$, we have with probability at least $1 - O(d^{-3})$, all $f$ and all $i$ satisfy

$$\left| \frac{1}{T} \sum_{x \in T} x_i f(x) - \mathop{\mathbb{E}}_{x \sim \mathcal{D}} x_i f(x) \right| \leq 1/d.$$

$\square$

**Claim C.12** (Accurate partitioning of (randomized) sets). *Let $\mathcal{D}$ be a distribution over $\mathbb{R}^d \times \{-1, 1\}$ such that the $\mathbb{R}^d$-marginal is isotropic log-concave. Let $\varepsilon > d^{-1/7}$. Let $\mathcal{F}$ be the set of eight (possibly randomized) Boolean functions over $\mathbb{R}^d$ given by* $\mathrm{RobustIndicator}_i$ *and* $\mathrm{ErrorIndicator}_i(x) := \mathbb{1}[y \neq h_i(x)]$ *for $i \in [4]$. Let $w_1, \dots, w_4$ be the weights returned by* COMPUTEROUNDINGTHRESH-OLDS *and $J_1, \dots, J_4$ be the intervals generated by* COMPUTECLASSIFIER. *Then with probability at least $1 - O(\delta)$, the following holds for all $f \in \mathcal{F}$ and all $i \in [4]$:*

$$\Pr_{x \sim \mathcal{D}}[\langle x, u \rangle \in J_i \wedge f(x) = 1] = \Pr_{x \sim \mathcal{D}}[f(x) = 1] \cdot w_i \pm O(\varepsilon).$$

*Proof.* First we will claim that each iteration succeeds with probability $1 - 1/\mathrm{poly}(n)$. Consider only the functions $f \in \mathcal{F}$ such that $\Pr[f(x) = 1] \geq \varepsilon$; for the others, the statement holds trivially. COMPUTECLASSIFIER estimates the means $\mu_i, \mu_i'; i \in [4]$ of the functions from its sample of size $10d^3$, then sets $P$ to be a linear subspace orthogonal to all of these. By Claim C.11 we have that with probability $1 - O(d^{-3})$ the mean-accuracy assumption of Claim C.10 is satisfied. Thus, by Claim C.10, the assumptions of Claim C.6 are satisfied with the following parameters:

- $\beta_1 = O(\sqrt{1/d})$

- $\beta_2 = O(d^{-1/6} \ln d)$

- $\beta_3 = \varepsilon - O(1/d)$.

COMPUTECLASSIFIER then generates a uniform random unit vector $u \in P$ and four intervals $J_1 \dots J_4$ of Gaussian volume $w_1, \dots, w_4$. By Claim C.6, with high probability over $u$, the following holds for each $J_i$:

$$\Pr_{x \sim \mathcal{D}|f(x)=1}[\langle x, u \rangle \in J_i] = w_i \pm O(\beta_1/\beta_3 + \beta_2 + \varepsilon) = w_i \pm O(\varepsilon);$$

taking the conjunction with the event that $f(x) = 1$ gives us the desired

$$\Pr_{x \sim \mathcal{D}}[\langle x, u \rangle \in J_i \wedge f(x) = 1] = \Pr_{x \sim \mathcal{D}}[f(x) = 1] \cdot (w_i \pm O(\varepsilon)) \leq \Pr_{x \sim \mathcal{D}}[f(x) = 1] \cdot w_i \pm O(\varepsilon).$$

The failure probability is

$$\frac{(1 + \beta_1/\beta_3) \ln(1/\varepsilon)}{\varepsilon^3 \beta_3 (1 - 2\beta_2 - 2\beta_1)} \cdot \exp\left(-\Omega\left(\frac{\varepsilon^4 \beta_3 (1 - 2\beta_2 - 2\beta_1)d}{\ln(1/\varepsilon)(1 + \beta_1/\beta_3)}\right)\right)$$

$$\frac{(1 + o(1)) \ln(1/\varepsilon)}{\varepsilon^4 (1 - o(1))} \cdot \exp\left(-\Omega\left(\frac{\varepsilon^5 (1 - o(1))d}{\ln(1/\varepsilon)(1 + o(1))}\right)\right)$$

$$\frac{\ln(1/\varepsilon)}{\varepsilon^4} \cdot \exp\left(-\Omega\left(\frac{\varepsilon^5 d}{\ln(1/\varepsilon)}\right)\right)$$

$$1/\varepsilon^5 \cdot \exp\left(-\Omega(\varepsilon^6 d)\right)$$

By our assumption that $\varepsilon > \Omega(d^{-1/7})$, this term is dominated by $\exp(-\Omega(d^{1/7}))$. The total failure probability is dominated by the $O(d^{-3})$ probability of failure for the mean estimation. Thus, we have that with probability $\geq 1 - 1/\mathrm{poly}(d)$, no bad tail events occur, and the guarantee holds for all $f \in \mathcal{F}$ and $i \in [4]$.

The following boosting argument is relevant only when $\delta$ is smaller than this $\approx d^{-3}$ failure probability. In each iteration we test the accuracy of the partition by comparing

$$\Pr[\langle x, u \rangle \in J_i \wedge f(x) = 1] \quad \text{to} \quad \Pr[f(x) = 1] \cdot w_i$$

for each function $f$. It suffices for these estimates to be accurate up to $\varepsilon$ additive error. By a Chernoff bound, for a test set of size $\Omega(\log(1/\delta)/\varepsilon^2)$, all estimates are $\varepsilon$-accurate with probability at least $1 - \delta^2$, so that after union bounding over all iterations the estimates are still accurate with probability at least $1 - \delta$. Since $d \geq 2$, the probability that a good partition is not found in any of the $\log(1/\delta)$ independent attempts is at most $2^{-\log(1/\delta)} = \delta$. Thus, the total failure probability is at most $2\delta$. $\quad\square$

## C.4 Correctness and complexity of Algorithm 5.

Now we will finish the analysis of COMPUTECLASSIFIER, restated here for convenience.

**Theorem 3.4** (Correctness of Algorithm 5, COMPUTECLASSIFIER). *Let $\varepsilon \geq d^{-1/7}$ and $\delta, r > 0$. Let $\hat{\phi}$ be an $\varepsilon$-accurate local noise sensitivity approximator for degree-$k$ PTFs over $\mathbb{R}^d$. Let $\mathcal{D}$ be a distribution over $\mathbb{R}^d \times \{-1, 1\}$ such that the $\mathbb{R}^d$-marginal is isotropic and log-concave. Let $p : \mathbb{R}^d \to \mathbb{R}$ be a polynomial of degree at most $k$ satisfying*

$$\mathop{\mathbb{E}}_{t \sim [-1,1]} \left[ \mathop{\mathbb{E}}_{(x,y) \sim \mathcal{D}} [\phi_{\mathrm{sign}(p-t), 10r}(x)] \right] \leq 100r + O(\varepsilon), \tag{5}$$

$$\Pr_{(x,y) \sim \mathcal{D}} \left[ \mathop{\mathbb{E}}_{t \sim [-1,1]} [\phi_{\mathrm{sign}(p(x)-t), 10r}(x)] \geq 0.7 \right] \leq O(\varepsilon). \tag{6}$$

*Then the algorithm COMPUTECLASSIFIER$(\mathcal{D}, p, r, \varepsilon, \delta)$, given sample access to $\mathcal{D}$ and query access to $\hat{\phi}$, outputs a hypothesis $h : \mathbb{R}^d \to \{-1, 1\}$ such that the following properties hold with probability at least $1 - O(\delta)$:*

$$\mathrm{err}_{\mathcal{D}}(h) \leq \mathop{\mathbb{E}}_{t \sim [-1,1]} [\mathrm{err}_{\mathcal{D}}(\mathrm{sign}(p-t))] + O(\varepsilon)$$

$$\mathrm{Boundary}_{\mathcal{D}, r}(h) \leq O(r + \varepsilon).$$

*The running time and number of queries to $\hat{\phi}$ are $\mathrm{poly}(d^k \cdot 1/\varepsilon \cdot \log(1/\delta))$.*

*Proof of Theorem 3.4.* By Theorem C.1 and the triangle inequality, we have that all $i \in [4]$ satisfy

(i) $\Pr_{(x,y) \sim \mathcal{D}}[h_i(x) \neq y] \leq \mathrm{err}_{\mathcal{D}}(\mathrm{sign}(p - t_i)) + \widehat{\mathrm{iso}}_{\mathcal{D}, 10r}(\mathrm{sign}(p - t_i), 0.8)$

(ii) $\mathbb{E}_{x \sim \mathcal{D}}[\mathrm{RobustIndicator}_i(x)] \geq 1 - O(\widehat{\mathrm{NS}}_{\mathcal{D}, 10r}(\mathrm{sign}(p - t_i))$

(iii) For every $x$ such that $\mathrm{RobustIndicator}_i(x) = 1$ and $x' : \|x - x'\| \leq r$, we have $h_i(x') = h_i(x)$.

First we will handle the error condition. By Theorem C.2, we have that with probability $1 - O(\delta)$,

$$\sum_{i \in [4]} w_i \cdot (\mathrm{err}_{\mathcal{D}}(\mathrm{sign}(p - t_i) + \widehat{\mathrm{iso}}_{\mathcal{D}, 10r}(\mathrm{sign}(p - t_i), 0.8))$$

$$\leq \mathop{\mathbb{E}}_{t \sim [-1,1]} [\mathrm{err}_{\mathcal{D}}(\mathrm{sign}(p - t))] + O(\varepsilon).$$

Thus, we have

$$\sum_{i \in [4]} w_i \cdot \Pr_{(x,y) \sim \mathcal{D}}[h_i(x) \neq y] \leq \mathop{\mathbb{E}}_{t \sim [-1,1]} [\mathrm{err}_{\mathcal{D}}(\mathrm{sign}(p - t))] + O(\varepsilon).$$

By Claim C.12, we then have with probability $\geq 1 - O(\delta)$,

$$\sum_{i \in [4]} \Pr_{(x,y) \sim \mathcal{D}}[h_i(x) \neq y \wedge \langle x, u^\star \rangle \in J_i] \leq \mathop{\mathbb{E}}_{t \sim [-1,1]} [\mathrm{err}_{\mathcal{D}}(\mathrm{sign}(p - t))] + O(\varepsilon).$$

Therefore, applying the definition of $h$, we have

$$\mathrm{err}_{\mathcal{D}}(h) \leq \mathbb{E}_{t \sim [-1,1]}[\mathrm{err}_{\mathcal{D}}(\mathrm{sign}(p - t))] + O(\varepsilon).$$

Now we analyze the robustness condition. By Theorem C.2, we have that with probability $\geq 1 - O(\delta)$,

$$\sum_{i \in [4]} w_i \cdot \widehat{\mathrm{NS}}_{\mathcal{D}, 10r}(\mathrm{sign}(p - t_i)) \leq O(r + \varepsilon).$$

By a Markov bound and the definitions of RobustIndicator and $\widehat{\mathrm{NS}}$, we have the following for each $i$:

$$\mathbb{E}[\mathrm{RobustIndicator}_i(x)] = \Pr[\hat{\phi}_{\mathrm{sign}(p-t_i)}(x) \leq 0.1] = 1 - \Pr[\hat{\phi}_{\mathrm{sign}(p-t_i)}(x) > 0.9]$$
$$= 1 - \Pr\left[\hat{\phi}(x) > 0.9 / \widehat{\mathrm{NS}}(\mathrm{sign}(p - t_i)) \cdot \mathbb{E}[\hat{\phi}(x)]\right]$$
$$\geq 1 - \tfrac{10}{9}\widehat{\mathrm{NS}}(\mathrm{sign}(p - t_i)).$$

Thus we have

$$\sum_{i \in [4]} w_i \cdot \mathbb{E}_{\mathcal{D}}[\mathrm{RobustIndicator}_i(x)] \geq \sum_{i \in [4]} w_i \cdot (1 - \tfrac{10}{9}\mathbb{E}_{\mathcal{D}}[\hat{\phi}_{\mathrm{sign}(p-t_i), 10r}(x)])$$
$$\geq 1 - \sum_{i \in [4]} w_i \cdot \tfrac{10}{9}\mathbb{E}_{\mathcal{D}}[\hat{\phi}_{\mathrm{sign}(p-t_i), 10r}(x)]$$
$$\geq 1 - O(r + \varepsilon).$$

By Claim C.12 we then have with probability $\geq 1 - \delta$,

$$\sum_{i \in [4]} \mathbb{E}_{x \sim \mathcal{D}}[\mathrm{RobustIndicator}_i(x) \cdot \mathbb{1}[\langle x, u^\star \rangle \in J_i]] \geq 1 - O(r + \varepsilon).$$

We now claim that for each $J_i$, all $x$ with $\mathrm{RobustIndicator}_i(x) = 1$ such that $\langle x, u^\star \rangle$ is at least $r$ away from the interval boundary satisfy the adversarial robustness condition

$$\forall x' : \|x - x'\|_2 \leq r \text{ and } h(x) = h(x').$$

Since $x$ is $r$ away from the interval boundary, all $x'$ such that $\|x - x'\|_2 \leq r$ satisfy $\langle x', u^\star \rangle \in J_i$, so they are also labeled by $h_i$. By Theorem C.1, all $x$ with $\mathrm{RobustIndicator}_i(x) = 1$ satisfy the robustness condition in $h_i$, so, since they are also $r$ away from the boundary, they satisfy the robustness condition in $h$. Thus we have

$$\mathrm{Boundary}_{\mathcal{D}, r}(h) \geq \sum_{i \in [4]} \left( \mathbb{E}_{x \sim \mathcal{D}}[\mathrm{RobustIndicator}_i(x) \cdot \mathbb{1}[\langle x, u^\star \rangle \in J_i]] \right)$$
$$- \Pr_{x \sim \mathcal{D}}[\langle x, u^\star \rangle \text{ within } r \text{ of an interval boundary}]$$
$$\geq 1 - O(r + \varepsilon) - O(r),$$

where the boundary probability is bounded by $O(r)$ due to Fact B.4. Thus when all subroutines are successful, the guarantees of the theorem hold. The total success probability is $\geq 1 - O(\delta)$ after union bounding the failure probabilities of all subroutines. This concludes the proof of correctness.

We now analyze the running time and query complexity to $\hat{\phi}$. COMPUTEROUNDINGTHRESHOLDS is called once. For $\log(1/\delta)$ iterations, it estimates $\mathrm{err}_t, \mathrm{NS}_t, \mathrm{iso}_t$ for each of $100/\varepsilon^2$ rounding thresholds $t$; this step makes $O(\log(1/\delta)/\varepsilon^2)$ queries to $\hat{\phi}$ and takes $\mathrm{poly}(d^k \cdot 1/\varepsilon \cdot \log(1/\delta))$ additional time, due to the evaluations of the degree-$k$ polynomial $p$. Then it solves a linear program of constant size for $(100/\varepsilon^2)^4$ iterations. The total running time and query complexity of COMPUTEROUNDINGTHRESHOLDS are dominated by the first term, which is $\mathrm{poly}(d^k \cdot 1/\varepsilon \cdot \log(1/\delta))$.

The rest of COMPUTECLASSIFIER repeats $\log(1/\delta)$ times and does the following each repetition:

a) evaluate $\mathbb{1}[h_i(x) \neq y]$ and $\mathrm{RobustIndicator}_i(x)$ for each $i \in [4]$ and $(x, y)$ in $M$,

b) obtain a basis orthogonal to the estimated mean vectors, a random unit vector in this space, and a partition of the real line into intervals

c) check accuracy of the partition with respect to $S_{\mathrm{test}}$.

Each evaluation of $\mathbb{1}[h_i(x) \neq y]$ makes one query to $\hat{\phi}$ and takes $d^{O(k)}$ time, as it simply calls ROBUSTNESSLCA, which makes one evaluation of a degree-$k$ PTF and one call to $\hat{\phi}$. Each evaluation of $\mathrm{RobustIndicator}_i(x)$ makes one query to $\hat{\phi}$ and takes $O(1)$ additional time. Thus items (a) and (c) take $d^{O(k)}$ time and queries. For item (b), obtaining the basis takes $\mathrm{poly}(d)$ time by Gaussian elimination, and approximating the interval boundaries can be done in $\mathrm{poly}(1/\varepsilon)$ time by using a numerical $\varepsilon$-approximation to the error function[10]

$$\mathrm{erf}(t) = \frac{1}{\sqrt{\pi}} \int_0^t e^{-t^2/2}.$$

Overall, the total time and query complexity of COMPUTECLASSIFIER is $\mathrm{poly}(d^k \cdot 1/\varepsilon \cdot \log(1/\delta))$, as desired. □

## D  Verifiable robustness

In this section we prove that under complexity assumptions, the robustness guarantee of our learning algorithm can be made efficiently verifiable as discussed in Section 1.3. The verifier certifies that a point satisfies the robustness condition. We formally state this result:

**Corollary D.1** (Deterministic verifiability). *If $\mathsf{P} = \mathsf{BPP}$, then there is a learning algorithm $\mathcal{B}$ that, given access to labeled samples from a subgaussian isotropic log-concave distribution, runs in time $d^{\tilde{O}(1/\varepsilon^2)} \cdot \log(1/\delta)$ and produces a hypothesis $h$ with the following guarantees:*

- *Agnostic approximation: With probability at least $1 - O(\delta)$, $\mathrm{Pr}_{(x,y)\sim\mathcal{D}}[h(x) \neq y] \leq \mathsf{opt} + O(\varepsilon)$, where $\mathsf{opt}$ is the misclassification error of the best halfspace.*

- *Verifiable robustness: There is a verifier that runs in time $d^{\tilde{O}(1/\varepsilon^2)}$ that takes as input a circuit $g$ and a point $x \in \mathbb{R}^d$ that always rejects if*

$$\exists z : \|z\|_2 \leq r \text{ and } g(x) \neq g(x+z).$$

*If $g = h$, then with probability at least $1 - O(\delta)$ over the randomness of $\mathcal{B}$, the verifier accepts with probability at least $1 - O(r + \varepsilon)$ over $x \sim D$.*

We use the following fact:

**Fact D.2** (Derandomized estimation of $\hat{\phi}$). *If $\mathsf{P} = \mathsf{BPP}$, then there exists a deterministic algorithm running in time $d^{O(k)} \cdot \mathrm{poly}(1/\varepsilon)$ that takes as input a degree-$k$ PTF $f$, a radius $r$, and an input $x \in \mathbb{R}^d$ and outputs an estimate $\hat{\phi}_{f,10r}(x)$ such that $\hat{\phi}_{f,10r}(x) = \phi_{f,10r}(x) \pm \varepsilon$.*

*Proof.* Observe that there is a randomized algorithm running in $d^{O(k)}/\varepsilon^2$ time that takes as input a threshold $t$ and decides if $\phi_f(x) \geq t$, succeeding with probability $\geq 2/3$ whenever $\phi_f(x) > t + \varepsilon$ or $\phi_f(x) < t - \varepsilon$. This is the algorithm that samples $O(1/\varepsilon^2)$ points $z$ from $\mathcal{N}(0, I_d)$ and evaluates $f(x + 10rz)$ on each of them to estimate $\phi$; its analysis is a Chernoff bound. If $\mathsf{P} = \mathsf{BPP}$, then there exists a deterministic algorithm for this decision problem also running in time $d^{O(k)} \cdot \mathrm{poly}(1/\varepsilon)$. Since $\phi_f(x) \in [0, 1]$, we can binary search with $\log(1/\varepsilon)$ iterations to find a $t$ such that $\phi_f(x) \in t \pm \varepsilon$. □

The learning algorithm is simply ROBUSTLEARN, augmented to provide some extra information, with the estimates of $\hat{\phi}$ provided by the deterministic estimator. The verifier checks that the hypothesis matches a "template;" this will prove that the unknown circuit is in fact of the form that ROBUSTLEARN is supposed to return, and for which our correctness analysis holds. Since for any hypothesis matching the template, any point $x$ for which $\phi(x) \leq 0.1$ satisfies the robustness condition, the verifier will then deterministically estimate $\phi(x)$.

We define the template below:

---

[10]Such numerical approximations are standard and take $\mathrm{poly}(1/\varepsilon)$ time. Alternatively, one could use a randomized algorithm that takes $\mathrm{poly}(1/\varepsilon)$ samples from the gaussian and uses them to approximate the relevant values $w_i$. Note that we presented the $w_i$'s as exact for conciseness, but since Claim C.12 already guarantees only an $O(\varepsilon)$-accurate partition, we see that an additional $\varepsilon$ error in the estimation of the $w_i$'s is asymptotically irrelevant.

**Lemma D.3** (Hypothesis template). *Fix a set $B$ of basis functions for the set of degree-$k$ polymomials over $\mathbb{R}^d$. There is an algorithm* COMPILE *that takes as input a $|B|$-length vector $\vec{v}$ of real-valued coefficients, real numbers $t_1, \ldots, t_4$, a unit vector $u \in \mathbb{R}^d$, and real numbers $c_1 < c_2 < c_3$. For each $i \in [4]$, let the PTF $h_i$ be defined*

$$h_i(x) = \text{sign}\left(\left(\sum_{b \in B} v_b \cdot b(x)\right) - t_i\right).$$

*Let the intervals $J_1, \ldots, J_4$ be the partition of the real line induced by $c_1, c_2, c_3$. The algorithm outputs a circuit computing the hypothesis*

$$h(x) = \sum_{i \in 4} \text{ROBUSTNESSLCA}(x, h_i, r) \cdot \mathbb{1}[\langle x, u \rangle \in J_i].$$

*It also outputs each of the PTFs $h_1, \ldots, h_4$. The running time is $d^{O(k)}$.*

*Proof.* Observe that there is a deterministic algorithm running in $d^{O(k)}$ time that takes as input all of the given parameters and the input $x$, and evaluates $h(x)$, using the deterministic implementation of $\hat{\phi}$ in ROBUSTNESSLCA. Therefore there is a circuit of size $d^{O(k)}$ with the same behavior. The algorithm COMPILE takes this circuit and hardcodes all the input parameters except for $x$, then outputs the resulting circuit. This takes time $d^{O(k)}$ and the output is a circuit that takes $x$ as input and evaluates $h(x)$. By the same argument, in $d^{O(k)}$ time COMPILE can also output the PTFs $h_1, \ldots, h_4$. $\qquad\square$

We include pseudocode of the verifier below. For brevity we will refer to the package of data taken as input by COMPILE as data.

---

**Algorithm 6** VERIFY$(g, x, r, \varepsilon, \text{data})$:

1: **Input:** circuit $g$, point $x \in \mathbb{R}^n$, robustness radius $r$, error tolerance $\varepsilon$, hypothesis parameters data
2: $h, h_1, h_2, h_3, h_4 \leftarrow$ COMPILE(data)
3: **if** $g \neq h$ **then return** reject.
4: Let $u$ be the unit vector in data and $J_1, \ldots, J_4$ be the partition induced by the interval boundaries $c_1, c_2, c_3$ in data.
5: Let $i$ be the interval such that $\langle x, u \rangle \in J_i$.
6: **if** $\hat{\phi}_{h_i, 10r}(x) > 0.1 - \varepsilon$ **then return** reject.
7: **if** $|\langle x, u \rangle - c_j| \leq r$ for any of the interval boundaries $c_1, c_2, c_3$ in data **then return** reject.
8: **return** accept.

---

*Proof of Corollary D.1.* The learning algorithm $\mathcal{B}$ is ROBUSTLEARN, but with the following modification: it records its parameters in a data package, outputs COMPILE(data) as its final hypothesis, and outputs data as well. By the guarantees of ROBUSTLEARN, with probability at least $1 - O(\delta)$, $h$ satisfies the agnostic approximation guarantee. When VERIFY is called on the output $h$ of $\mathcal{B}$, since it is the output of COMPILE, it always passes the first check. By inspecting the proof of Theorem C.1, we see that in fact robustness holds for all $x$ such that $\hat{\phi}_{h_i}(x) \leq 0.1 - \varepsilon$ and $x$ is at least $r$ distance from each of the interval boundaries. By Fact D.2, we have $|\hat{\phi}_{h_i}(x) - \phi_{h_i}(x)| \leq \varepsilon$, thus the verifier rejects all points such that $\phi_{h_i}(x) > 0.1$. Since it also rejects if $x$ is within $r$ of a boundary, all accepted points satisfy the robustness condition. Furthermore, by inspecting the proofs of Theorem 3.4 and Theorem C.1, we see that whenever ROBUSTLEARN succeeds (probability $1 - O(\delta)$), we have that at least $1 - O(r + \varepsilon)$ fraction of points $x \sim \mathcal{D}$ satisfy $\hat{\phi}_{h_i}(x) \leq 0.1 - \varepsilon$ and are at least $r$ away from any boundary, and are thus verifiably robust. Thus, the verifiable robustness guarantee holds.

The running time of COMPILE for PTFs of degree $\tilde{O}(1/\varepsilon^2)$ is $d^{\tilde{O}(1/\varepsilon^2)}$ (Lemma D.3). The running time of VERIFY is this plus the running time of the deterministic estimator for $\hat{\phi}$, which is also $d^{\tilde{O}(1/\varepsilon^2)}$ (Fact D.2). Thus the total running time is $d^{\tilde{O}(1/\varepsilon^2)}$. $\qquad\square$

**Remark D.4.** We note that without the assumption that $\mathsf{P} = \mathsf{BPP}$, there is a randomized analogue of Corollary D.1 where the user compiles the final hypothesis, rather than verifying the one provided by ROBUSTLEARN. Rather than outputting a circuit $h$, ROBUSTLEARN can just output data, and the user can COMPILE it with a randomized implementation of ROBUSTNESSLCA in time $d^{\tilde{O}(1/\varepsilon^2)} \cdot \log(1/\delta)$, in which case the soundness of the verifier holds with probability $1 - \delta$ for the compiled hypothesis.

# E    Uniform convergence claims

Finally, we will need to following observation about the uniform convergence fo empirical approximations of local noise sensitivity $\phi$. First, we need the following fact:

**Fact E.1** ([VW97], also see lecture notes [Duc23]). *A function class $\mathcal{C}$ of VC dimension $\Delta$ and every distribution $D_0$, there is an $\varepsilon$-cover $H$ of $\mathcal{C}$ of size at most $\beta := (O(1)/\varepsilon)^{O(\Delta)}$. I.e. $H$ is a discrete subset of $\mathcal{C}$ of size $\beta$ and for every $f$ in $\mathcal{C}$ we have $h$ in $H$ for which*

$$\Pr_{x \sim D_0}[f(x) \neq h(x)] \leq \varepsilon,$$

**Claim E.2.** *Let $\mathcal{C}$ be the class of degree-$k$ PTFs over $\mathbb{R}^d$, let $D$ be a probability distribution over $\mathbb{R}^d$ and let $\eta \in [0,1]$ be fixed. Then, for some constant $C$, if $S$ is a collection of $(d^k/\varepsilon)^C \log 1/\delta$ i.i.d. examples from $D$, then with probability at least $1 - O(\delta)$,*

$$\max_{f \in \mathcal{C}} \left| \mathbb{E}_{x \sim S}[\phi_{f,\eta}(x)] - \mathbb{E}_{x \sim D}[\phi_{f,\eta}(x)] \right| \leq O(\varepsilon) \tag{36}$$

*Proof.* We now use Fact E.1. For us, $\mathcal{C}$ is the class of degree-$k$ PTFs, and we have $\Delta = d^{O(k)}$. Taking the distribution $D_0$ to be an equal-weight mixture of (i) the distribution $D$ in the premise of this claim and (ii) the convolution of $D$ with the normal distribution $\mathcal{N}(0, \eta I_d)$ we see that for every $f$ in $\mathcal{C}$ there is $h$ in $H$ for which[11]

$$\Pr_{x \sim D}[f(x) \neq h(x)] \leq 2\varepsilon, \tag{37}$$

$$\Pr_{\substack{x \sim D \\ z \sim \mathcal{N}(0, I_d)}}[f(x + \eta z) \neq h(x + \eta z)] \leq 2\varepsilon, \tag{38}$$

which via the definition of $\phi$ and the triangle inequality implies that

$$\left| \mathbb{E}_{x \sim D}[\phi_{f,\eta}(x)] - \mathbb{E}_{x \sim D}[\phi_{h,\eta}(x)] \right| \leq \mathbb{E}_{x \sim D}\left[ \left| \phi_{f,\eta}(x) - \phi_{h,\eta}(x) \right| \right]$$

$$\leq 2\,\mathbb{E}_{x \sim D}[|f(x) - h(x)|] + \mathbb{E}_{\substack{x \sim D \\ z \sim \mathcal{N}(0, I_d)}}[|f(x + \eta z) - h(x + \eta z)|] \leq O(\varepsilon). \tag{39}$$

By the standard Hoeffding bound, since $\phi$ and $\xi$ are always in $[0,1]$, with probability $1 - \delta/2$ for every $h$ in $H$ we have

$$\left| \mathbb{E}_{x \sim S}[\phi_{h,\eta}(x)] - \mathbb{E}_{x \sim D}[\phi_{h,\eta}(x)] \right| \leq \frac{1}{\sqrt{|S|}} O(\log(|H|/\delta)) \leq \varepsilon, \tag{40}$$

where the last step for both inequalities follows by substituting $|H|$, $|S|$, $\Delta$ and taking $C$ to be a sufficiently large absolute constant. We also know that with for a sufficiently large absolute constant $C$, with probability $1 - \delta$ the set $S$ satisfies Fact E.6, which means that:

$$\sup_{f_1, f_2 \in \mathcal{C}} |\Pr_{x \sim S}[f_1(x) \neq f_2(x)] - \Pr_{x \sim D}[f_1(x) \neq f_2(x)]| \leq O(\varepsilon). \tag{41}$$

From the above, we can also conclude that with probability $1 - \delta$ over $S$ for every pair $f_1, f_2$ in $\mathcal{C}$ we have the following:

$$\left| \Pr_{\substack{x \sim S \\ z \sim \mathcal{N}(0, I_d)}}[f_1(x + \eta z) \neq f_2(x + \eta z)] - \Pr_{\substack{x \sim D \\ z \sim \mathcal{N}(0, I_d)}}[f_1(x + \eta z) \neq f_2(x + \eta z)] \right| \leq$$

$$\mathbb{E}_{z \sim \mathcal{N}(0, I_d)} \underbrace{\left| \Pr_{x \sim S}[f_1(x + \eta z) \neq f_2(x + \eta z)] - \Pr_{x \sim D}[f_1(x + \eta z) \neq f_2(x + \eta z)] \right|}_{\substack{\leq O(\varepsilon) \text{ by defining } f_3(x) := f_1(x + \eta z),\ f_4(x) := f_2(x + \eta z), \\ \text{observing } f_3 \text{ and } f_4 \text{ are also degree-}k \text{ PTFs and using Equation (41)}}} \leq O(\varepsilon)$$

$$\tag{42}$$

---

[11] Note that the error over $D_0$ is the average of the two errors in the inequalities below.

Using the definition of $\phi$, the triangle inequality, and Equations 41 and 42 we also see that

$$
\begin{aligned}
|\mathbb{E}_{x \sim S}[\phi_{h,\eta}(x)] - \mathbb{E}_{x \sim S}[\phi_{f,\eta}(x)]| \leq \\
2 \Pr_{x \sim S}[h(x) \neq f(x)] + \Pr_{\substack{x \sim S \\ z \sim \mathcal{N}(0,I_d)}}[h(x+\eta z) \neq f(x+\eta z)] \leq \\
2 \Pr_{x \sim D}[h(x) \neq f(x)] + \Pr_{\substack{x \sim D \\ z \sim \mathcal{N}(0,I_d)}}[h(x+\eta z) \neq f(x+\eta z)] \leq O(\varepsilon), \quad (43)
\end{aligned}
$$

where in the last step we substituted Equation (37) and Equation (38). Finally, we put all the inequalities together. By combining the triangle inequality with Equations 39, 40 and 43 we derive Equation (36). $\qquad \square$

**Claim E.3.** *Let $\mathcal{C}$ be the class of degree-$k$ PTFs over $\mathbb{R}^d$, let $D$ be a probability distribution over $\mathbb{R}^d$ and let $\eta \in [0,1]$ be fixed. Then, for some constant $C$, if $S$ is a collection of $(d^k/\varepsilon)^C \log 1/\delta$ i.i.d. examples from $D$, then with probability at least $1 - O(\delta)$, for every polynomial $p$ it holds that*

$$
\Pr_{x \sim D}\left[\mathbb{E}_{t \in [-1,1]}[\phi_{\mathrm{sign}(p(x)-t),\eta}(x)] \geq 0.7\right] \leq \Pr_{x \sim D}[\mathbb{E}_{t \in [-1,1]}[\phi_{\mathrm{sign}(p-t),\eta}(x)] \geq 0.67] + O(\varepsilon) \quad (44)
$$

*Proof.* Let $\mathcal{C}^{\mathrm{clipped}}$ be the class of degree-$k$ clipped polynomials, i.e. functions $p^{\mathrm{clipped}} : \mathbb{R}^d \to [-1,1]$ of the form

$$
p^{\mathrm{clipped}}(x) = \begin{cases} 1 & \text{if } p(x) \geq 1 \\ -1 & \text{if } p(x) \leq -1 \\ p(x) & \text{otherwise} \end{cases}
$$

where $p$ is a degree-$k$ polynomial. We first argue the following

**Observation E.4.** *For any distribution $D_0$ over $\mathbb{R}^d$, there is a subset $H$ of $\mathcal{C}^{\mathrm{clipped}}$ of size $(O(1)/\varepsilon)^{O(d^k/\varepsilon)}$, i.e. such that for every $p^{\mathrm{clipped}}$ in $\mathcal{C}^{\mathrm{clipped}}$ we have*

$$
\min_{h \in H}[\mathbb{E}_{x \sim D_0}[|p^{\mathrm{clipped}}(x) - h(x)|]] \leq O(\varepsilon). \quad (45)
$$

*Proof.* We form $H$ by considering an $\varepsilon^2$-net $H_{\mathrm{PTF}}^{\varepsilon^2}$ for degree-$2k$ PTFs, and taking $H$ to consist of functions of the form

$$
h(x) = \sum_{\tau \in \{-1,-1+\varepsilon,-1+2\varepsilon,\ldots,+1\}} \tau \cdot f_\tau(x),
$$

where each $f_\tau(x)$ is some function in $f_\tau(x)$. By Fact E.1, we see that $H_{\mathrm{PTF}}^{\varepsilon^2}$ has size $(O(1)/\varepsilon)^{O(d^k)}$, and therefore (by a counting argument) the size of $H$ defined above is indeed $(O(1)/\varepsilon)^{O(d^k/\varepsilon)}$.

We can write for every $x$ and $p^{\mathrm{clipped}}$ in $\mathcal{C}^{\mathrm{clipped}}$ the following inequality:

$$
\sum_{\tau \in \{-1,-1+\varepsilon,-1+2\varepsilon,\ldots,+1\}} \tau \cdot \mathbb{1}[p^{\mathrm{clipped}}(x) \in (\tau, \tau+\varepsilon]] \leq p^{\mathrm{clipped}}(x) \leq
$$

$$
\sum_{\tau \in \{-1,-1+\varepsilon,-1+2\varepsilon,\ldots,+1\}} (\tau+\varepsilon) \cdot \mathbb{1}[p^{\mathrm{clipped}}(x) \in (\tau, \tau+\varepsilon]], \quad (46)
$$

we observe that each indicator $\mathbb{1}[p^{\mathrm{clipped}}(x) \in (\tau, \tau+\varepsilon]]$ equals to a degree-$2k$ polynomial threshold function, and therefore there is some $f_\tau$ in $H_{\mathrm{PTF}}^{\varepsilon^2}$ for which

$$
\mathbb{E}_{x \sim D_0}\left|\mathbb{1}[p^{\mathrm{clipped}}(x) \in (\tau, \tau+\varepsilon]] - f_\tau\right| \leq O(\varepsilon^2),
$$

which substituted into Equation (46) allows us to conclude that

$$
\mathbb{E}_{x \sim D_0}\left|p^{\mathrm{clipped}}(x) - \sum_{\tau \in \{-1,-1+\varepsilon,-1+2\varepsilon,\ldots,+1\}} \tau \cdot f_\tau(x)\right| \leq O(\varepsilon),
$$

which concludes the proof that for $p^{\mathrm{clipped}}$ in $\mathcal{C}^{\mathrm{clipped}}$ we have $h : \mathbb{R}^d$ in $H$ for which Equation (45) holds.

We finish the proof of the observation, by resolving one last issue: as defined the set $H$ is not a subset of $\mathcal{C}^{\text{clipped}}$. However, if we consider the set $H'$ consisting of functions $f$ of the form

$$H' = \{f = \underset{g \in \mathcal{C}^{\text{clipped}}}{\text{argmin}}[\mathbb{E}_{x \sim D_0}[|g(x) - h(x)|]] : h \in H,\}$$

then we see by the triangle inequality that $H'$ still satisfies Equation (45), has a size of at most $|H|$ and $H'$ is a subset of $\mathcal{C}^{\text{clipped}}$ $\qquad\square$

We continue the proof of Claim E.3 We can wite:

$$\mathbb{E}_{t \in [-1,1]}[\phi_{\text{sign}(p(x)-t),\eta}(x)] = \Pr_{t \in [-1,1], z \sim \mathcal{N}(0,I_d)}[|\text{sign}(p(x)-t) \neq \text{sign}(p(x+\eta z)-t)|] =$$

$$\mathbb{E}_{z \sim \mathcal{N}(0,I_d)}\left[\frac{\left|p^{\text{clipped}}(x) - p^{\text{clipped}}(x+\eta z)\right|}{2}\right] := \phi_{p^{\text{clipped}},\eta}(x) \quad (47)$$

We will overload the notation for $\phi$ and define $\phi_{p^{\text{clipped}},\eta}(x)$ to be the expression above. We further define the following auxiliary quantity:

$$\xi_{p^{\text{clipped}},\eta}(x) = \begin{cases} 0 & \text{if } \phi_{p^{\text{clipped}},\eta}(x) \leq 0.67 \\ 1 & \text{if } \phi_{p^{\text{clipped}},\eta}(x) \geq 0.7 \\ \frac{100}{3}(\phi_{p^{\text{clipped}},\eta}(x) - 0.67) & \text{if } \phi_{f,\eta}(x) \in (0.67, 0.7) \end{cases}$$

By construction, the function satisfies the following properties for every $x$,

$$\mathbb{1}[\phi_{p^{\text{clipped}},\eta}(x) \geq 0.67] \leq \xi_{p^{\text{clipped}},\eta}(x) \leq \mathbb{1}[\phi_{p^{\text{clipped}},\eta}(x) \geq 0.7] \quad (48)$$

$$\left|\xi_{p_1^{\text{clipped}},\eta}(x) - \xi_{p_2^{\text{clipped}},\eta}(x)\right| \leq O(1) \cdot \left|\phi_{p_1^{\text{clipped}},\eta}(x) - \phi_{p_2^{\text{clipped}},\eta}(x)\right| \quad (49)$$

Taking the distribution $D_0$ to be an equal-weight mixture of (i) the distribution $D$ in the premise of this claim and (ii) the convolution of $D$ with the normal distribution $\mathcal{N}(0, \eta I_d)$ we see that for every $p^{\text{clipped}}$ in $\mathcal{C}^{\text{clipped}}$ there is $h$ in $H$ for which[12]

$$\Pr_{x \sim D}[|p^{\text{clipped}}(x) - h(x)|] \leq 2\varepsilon, \quad (50)$$

$$\Pr_{\substack{x \sim D \\ z \sim \mathcal{N}(0,I_d)}}[|p^{\text{clipped}}(x+\eta z) - h(x+\eta z)|] \leq 2\varepsilon, \quad (51)$$

which via the definition of $\phi$ for clipped polynomials and the triangle inequality implies that

$$\left|\mathbb{E}_{x \sim D}[\phi_{p^{\text{clipped}},\eta}(x)] - \mathbb{E}_{x \sim D}[\phi_{h,\eta}(x)]\right| \leq \mathbb{E}_{x \sim D}\left[\left|\phi_{p^{\text{clipped}},\eta}(x) - \phi_{h,\eta}(x)\right|\right]$$

$$\leq 2\,\mathbb{E}_{x \sim D}[|p^{\text{clipped}}(x) - h(x)|] + \mathbb{E}_{\substack{x \sim D \\ z \sim \mathcal{N}(0,I_d)}}[|p^{\text{clipped}}(x+\eta z) - h(x+\eta z)|] \leq O(\varepsilon). \quad (52)$$

Together with Equation (49), this implies that

$$\left|\mathbb{E}_{x \sim D}[\xi_{p^{\text{clipped}},\eta}(x)] - \mathbb{E}_{x \sim D}[\xi_{h,\eta}(x)]\right| \leq O(1) \cdot \mathbb{E}_{x \sim D}\left[\left|\phi_{p^{\text{clipped}},\eta}(x) - \phi_{h,\eta}(x)\right|\right] \leq O(\varepsilon). \quad (53)$$

By the standard Hoeffding bound, since $\phi$ and $\xi$ are always in $[0,1]$, with probability $1 - \delta/2$ for every $h$ in $H$ we have

$$\left|\mathbb{E}_{x \sim S}[\phi_{h,\eta}(x)] - \mathbb{E}_{x \sim D}[\phi_{h,\eta}(x)]\right| \leq \frac{1}{\sqrt{|S|}}O(\log(|H|/\delta)) \leq \varepsilon, \quad (54)$$

$$\left|\mathbb{E}_{x \sim S}[\xi_{h,\eta}(x)] - \mathbb{E}_{x \sim D}[\xi_{h,\eta}(x)]\right| \leq \frac{1}{\sqrt{|S|}}O(\log(|H|/\delta)) \leq \varepsilon, \quad (55)$$

where the last step for both inequalities follows by substituting $|H|$, $|S|$, $\Delta$ and taking $C$ to be a sufficiently large absolute constant.

---

[12]Note that the error over $D_0$ is the average of the two errors in the inequalities below.

We observe that Fact E.6 and Equation (46) together imply that[13]:

$$\sup_{p_1^{\text{clipped}}, p_2^{\text{clipped}} \in \mathcal{C}^{\text{clipped}}} \left| \mathbb{E}_{x \sim S}\left[\left|p_1^{\text{clipped}}(x) - p_2^{\text{clipped}}(x)\right|\right] - \mathbb{E}_{x \sim D}\left[\left|p_1^{\text{clipped}}(x) - p_2^{\text{clipped}}(x)\right|\right] \right| \leq O(\varepsilon).$$

(56)

From the above, we can also conclude that with probability $1 - \delta$ over $S$ for every pair $p_1^{\text{clipped}}, p_2^{\text{clipped}}$ in $\mathcal{C}^{\text{clipped}}$ we have the following:

$$\left| \mathbb{E}_{\substack{x \sim S \\ z \sim \mathcal{N}(0, I_d)}}\left[\left|p_1^{\text{clipped}}(x + \eta z) - p_1^{\text{clipped}}(x + \eta z)\right|\right] - \mathbb{E}_{\substack{x \sim D \\ z \sim \mathcal{N}(0, I_d)}}\left[\left|p_1^{\text{clipped}}(x + \eta z) - p_2^{\text{clipped}}(x + \eta z)\right|\right] \right| \leq$$

$$\mathbb{E}_{z \sim \mathcal{N}(0, I_d)} \underbrace{\left| \mathbb{E}_{x \sim S}\left[\left|p_1^{\text{clipped}}(x + \eta z) - p_2^{\text{clipped}}(x + \eta z)\right|\right] - \mathbb{E}_{x \sim D}\left[\left|p_1^{\text{clipped}}(x + \eta z) - p_2^{\text{clipped}}(x + \eta z)\right|\right] \right|}_{\leq O(\varepsilon) \text{ by defining } p_3^{\text{clipped}}(x) := p_1^{\text{clipped}}(x + \eta z), p_4^{\text{clipped}}(x) := p_2^{\text{clipped}}(x + \eta z),} \leq O(\varepsilon)$$
$$\text{and using Equation (56)}$$

(57)

We now come back to the setting of Equation (50). $p^{\text{clipped}}$ is a function in $\mathcal{C}^{\text{clipped}}$ and $h$ in $H$ satisfies Equation (50). Using the definition of $\phi$, the triangle inequality, and Equations 56 and 57 we also see that

$$\left| \mathbb{E}_{x \sim S}[\phi_{h, \eta}(x)] - \mathbb{E}_{x \sim S}[\phi_{p^{\text{clipped}}, \eta}(x)] \right| \leq$$
$$2 \Pr_{x \sim S}\left[\left|h(x) - p^{\text{clipped}}(x)\right|\right] + \Pr_{\substack{x \sim S \\ z \sim \mathcal{N}(0, I_d)}}\left[\left|h(x + \eta z) - p^{\text{clipped}}(x + \eta z)\right|\right] \leq$$
$$2 \Pr_{x \sim D}[h(x) \neq f(x)] + \Pr_{\substack{x \sim D \\ z \sim \mathcal{N}(0, I_d)}}[h(x + \eta z) \neq f(x + \eta z)] \leq O(\varepsilon), \quad (58)$$

where in the last step we substituted Equation (50) and Equation (51). By Equation 49, we see that

$$|\mathbb{E}_{x \sim S}[\xi_{h, \eta}(x)] - \mathbb{E}_{x \sim S}[\xi_{f, \eta}(x)]| \leq O(1) \cdot |\mathbb{E}_{x \sim S}[\phi_{h, \eta}(x)] - \mathbb{E}_{x \sim S}[\phi_{f, \eta}(x)]| \leq O(\varepsilon), \quad (59)$$

Analogously, by combining the triangle inequality with Equations 53, 55 and 59 we derive the following:

$$\max_{f \in \mathcal{C}} \left| \mathbb{E}_{x \sim S}[\xi_{f, \eta}(x)] - \mathbb{E}_{x \sim D}[\xi_{f, \eta}(x)] \right| \leq O(\varepsilon), \quad (60)$$

which together with Equation (48) and Equation (47) implies Equation (44).

$\square$

The following claims follow from VC theory (see e.g. [VDVW09] and the references therein):

**Fact E.5** (Generalization bound from VC dimension). *Let $\mathcal{C}$ be a concept class and $\mathcal{D}$ be a distribution over $\mathbb{R}^d$. For some sufficiently large absolute constant $C$, the following is true. With probability at least $1 - \delta$ over a sample $S$ of i.i.d. samples from $\mathcal{D}$, with $|S| \geq C \cdot VC(\mathcal{C}) \log(1/\delta)/\varepsilon^2$ the following holds:*

$$\sup_{f \in \mathcal{C}} |\text{err}_{\mathcal{D}}(f) - \widehat{\text{err}}_S(f)| \leq O(\varepsilon).$$

**Fact E.6.** *Let $\mathcal{C}$ be a concept class with VC dimension $\Delta$ and $D$ be a distribution over $\mathbb{R}^d$. For a sufficiently large absolute constant $C$, let $S$ be a collection of $\frac{C\Delta}{\varepsilon^2} \log(1/\delta)$ i.i.d. examples from $D$. Then, the following holds with probability at least $1 - \delta$*

$$\sup_{f_1, f_2 \in \mathcal{C}} |\Pr_{x \sim S}[f_1(x) \neq f_2(x)] - \Pr_{x \sim D}[f_1(x) \neq f_2(x)]| \leq O(\varepsilon). \quad (61)$$

---

[13]Equation (46) tells us that the difference $p_1 - p_2$ can be $\varepsilon$-approximated in $L_\infty$ norm by a function of the form $\sum_{\tau \in \{-2, -2+\varepsilon, -1+2\varepsilon, \ldots, +2\}} \tau \cdot f_\tau(x)$, where each $f_\tau$ is a degree-$2k$ PTF. Then, Fact E.6 together with triangle inequality tells us that the inequality above holds when $C$ is a sufficiently large absolute constant.

