# OpenReview forum: "Robust learning of halfspaces under log-concave marginals"
_NeurIPS.cc/2025/Conference — NeurIPS 2025 spotlight_

### Official Review · Reviewer_2jUf · 2025-06-01

**Clarity:** 4
**Significance:** 3
**Originality:** 3
**Rating:** 6
**Confidence:** 4

**Summary:**

This paper presents a novel and efficient algorithm for agnostically learning linear separators (halfspaces) under the assumption that the data's marginal distribution is isotropic and log-concave. The learned hypothesis is constructed from a Polynomial Threshold Function (PTF).

The paper makes two primary contributions:
1.  **Adversarial Robustness:** The algorithm produces a classifier that is provably robust against adversarial perturbations. This is formalized by guaranteeing a small "boundary volume" ($O(r+\epsilon)$ for a perturbation radius $r$), meaning that for most inputs, small perturbations do not cause a misclassification. This robustness is achieved while matching the optimal classification error ($opt+O(\epsilon)$) and the state-of-the-art sample complexity ($d^{\tilde{O}(1/\epsilon^2)}$) of the best *non-robust* learning algorithms for this problem.
2.  **Verifiable Robustness:** Under the standard complexity assumption $P=BPP$, the paper provides an efficient, deterministic verifier. This allows a user to certify that any specific input point is robustly classified by the learned hypothesis, addressing the practical problem of trusting models from untrusted sources.

**Questions:**

1.  Regarding verifiable robustness and the $P=BPP$ assumption:
    - The name for Corollary 1.5, "Deterministic robustness," is a bit confusing. Since the robustness guarantee for the hypothesis still holds probabilistically over the data distribution, would "Deterministic Verifiability" be a more accurate term? The key deterministic part seems to be the verifier itself, which can check any given point without randomness.
    - There appears to be a minor inconsistency in the presentation of Algorithm 6 (VERIFY). The input description on line 1 of the algorithm box mentions taking a point $x$ and an error tolerance $\epsilon$, but the function signature at the top of the box is `VERIFY(g, r, eps, data)`. Could the authors clarify the intended signature and inputs?
    - Are consecutive runs of the verifier on different points $x_1$ and $x_2$ independent? It seems they are in the $P=BPP$ setting, since the verifier is deterministic. Does this independence still hold for the randomized verifier mentioned in Remark D.4, or does the analysis change given that the hypothesis itself is constructed with randomness?

2.  The assumption $\epsilon \ge d^{-1/7}$ in Theorem 3.1 is a key technical condition. Could the authors add a brief remark explaining where in the proof this is necessary (it seems to be for the concentration of random projections in Claim C.6) and whether it is likely a limitation of the current analysis or potentially fundamental to the partitioning technique?

**Ethical Concerns:**

["NO or VERY MINOR ethics concerns only"]

**Final Justification:**

I find the technical parts of the paper well-written and only the introduction needs improvement, so the presentational issue seems minor to me (in contrast to another reviewer). The result is strong, so I maintain my positive score.

**Limitations:**

Yes

**Quality:**

4

**Strengths And Weaknesses:**

### Strengths

This paper presents a theoretical advance in robust learning, with clean technical insights and outstanding clarity.

- **Significance and Originality:** The paper addresses a core challenge in robustness: designing efficient learners that are provably robust. It gives the first algorithm to efficiently and robustly learn halfspaces in the agnostic setting under general log-concave distributions, matching the error guarantees and sample complexities of non-robust learners. Establishing _verifiable robustness_ is practically important for developing certifiably safe machine learning systems.

- **Technical Quality:** The technical contribution is deep, innovative, and very well-executed. The approach combines three key components, each introducing new ideas:

    1. Constrained Polynomial Regression: The learner fits a polynomial under convex constraints that enforce low noise sensitivity and isolation probability—critical for robustness.

    2. Structured Rounding and Partitioning: The real-valued polynomial is converted to a Boolean classifier using a randomized mixture (via Carathéodory’s theorem and LP) that preserves error and robustness. This mixture induces a partition with low boundary complexity.

    3. Local Correction: A final smoothing step turns a low-sensitivity classifier into one that is certifiably robust to adversarial perturbations.


- **Clarity:** The paper is exceptionally well-written. The introduction clearly motivates the problem and outlines the solution, while the technical sections provide a detailed and accessible analysis.

---
### Weaknesses

This is a very strong submission, but a few presentation details could be improved:

- The role of proper learning in the introduction could be clarified. Line 37 frames proper learning as a robustness benchmark, but it is not immediately clear that the proposed algorithm is _improper_. Since efficient proper agnostic learning remains an open problem, it would help to state explicitly and early that the paper achieves halfspace-level robustness without giving a proper learning algorithm.

- The informal versions of the main theorems (1.1 and 1.4) should reflect that the guarantees are probabilistic. Phrasing the results as holding “with high probability” would better align with the formal statements (Theorems 3.1 and 3.4). Adding forward references would also improve navigability.

---

> ### Author Rebuttal · Authors · 2025-07-29
>
> Thank you for suggesting that we mention that our algorithm is improper earlier in the introduction, add “with high probability” and forward references to the improper statements, and change the name “deterministic robustness” to “deterministic verifiability.” We will address these points in the final version of this paper.
>
> Thank you for pointing out as well that the function signature of Verify and the input description do not match. The intended inputs are $g, x, r, \epsilon, data$. We will fix this inaccuracy in the final version of this paper.
>
> Regarding the behavior of the randomized verifier, consecutive runs of the verifier on different points are not supposed to be independent; our guarantee is stronger than that. Rather, with probability $\ge 1 - \delta$ (over the choice of the samples of $N(0,I_d)$ used to implement the $\hat{\phi}$ estimator), soundness holds simultaneously for all points in $\mathbb{R}^d$.
>
> The condition $\epsilon > d^{-1/7}$ is indeed used for the partitioning step; it is because the failure probability of the concentration of projections has large powers of $\epsilon$. It may be possible with tighter analysis to decrease this power, but not below $\epsilon^4$, since that is the dependence given in [DHV ’06]. We would like to emphasize that when $\epsilon = d^{-1/7}$ the run-time is $2^{d^{O(1)}}$ which is exponential in $d$. Furthermore, for such small values of $\epsilon$, run-times of this form are believed to be necessary (even for regular improper learning) due to statistical query lower bounds we cite in this work. Since one of the main goals of this paper is to develop algorithms with run-times much faster than $2^{d^{O(1)}}$, we do not focus on the parameter regime where $
> \epsilon < d^{-1/7}$.

---

> > ### Comment · Reviewer_2jUf · 2025-08-08
> >
> > Thanks for the response. I'll maintain the positive score. I agree with another reviewer that the organization introduction can be slightly improved in the final version of the paper.

---

### Official Review · Reviewer_mppR · 2025-06-03

**Clarity:** 2
**Significance:** 2
**Originality:** 3
**Rating:** 4
**Confidence:** 3

**Summary:**

This paper considers the problem of agnostically learning halfspaces. That is given any Boolean function f over a well behaved (subgaussian and logconcave in this case) distribution D one wants to learn a hypothesis h so that Pr_{x sim D}(h(x) neq f(x)) <= OPT +eps, where OPT is the error of the best possible halfspace approximator. This can be done in d^O~(1/eps^2) samples using the L1 regression algorithm, which returns a hypothesis of a low degree PTF. On the other hand while the best-approximating halfspace is robust against adversarial examples (in the sense that for a random input x the probability that there is a nearby input y with |x-y| < r, with h(x) neq h(y) is small (O(r)), the PTF hypothesis may well not satisfy this. This paper develops techniques to learn hypothesis that both achieve the agnostic learning error guarantee but also satisfy this kind of adversarial robustness.

The basic idea seems to be the following:
The L1 regression algorithm uses linear programming to find a degree k (for k some O~(1/eps^2)) polynomial minimizing the empirical L1 loss E[|p(x)-f(x)|]. We would like to also add a constraint along the lines of E_x[ max_{|y-x|<r} |p(x)-p(y)| ] = O(r), which would ensure the kind of robustness that we want. Unfortunately, the maximization cannot be efficiently solved here. Instead, they replace this with a proxy E_{z ~ N(0,r*I)}[|p(x)-p(x+z)|], which can be efficiently approximated via sampling. It is not hard to show (using standard constructions) that there are polynomials that get OPT+eps error while simultaneously keeping this notion of noise sensitivity small.

By thresholding p, one obtains a boolean function with appropriate error and small "noise sensitivity" (i.e. probability that h(x) neq h(x+z) where x ~ D and z ~ N(0,r*I)). From here they say that a point x is isolated if Pr_z(h(x) neq h(x+z)) is large. It is then not hard to show that if two nearby points have different values of f, then at least one of them must be isolated. It can then be shown that flipping the value of h at all isolated points gives rise to a function with appropriately good adversarial robustness assuming that there are not too many points with modestly high noise sensitivity. To ensure that this doesn't flip too many points, one needs to add another constraint to the convex program to ensure that not too many points are isolated, however this is doable without affecting satisfiability.

One final issue is that the thresholding for p must be randomized and *on average* the total error is at most OPT+eps and *on average* the adversarial robustness is O(r+eps), but there is no guarantee that these are satisfied simultaneously. Fortunately, one can find only a few different thresholds that if averaged achieve this and by piecing the resulting functions together can obtain something appropriate.

Essentially all of the proofs were in the appendices so I did not read them in detail (NOTE TO THE PC: THIS IS A TERRIBLE SYSTEM THAT FORCES PEOPLE TO WRITE WORSE PAPERS THAT CANNOT BE CHECKED FOR CORRECTNESS ANYWAY WITHOUT READING THE EXTRA MATERIAL), but I did read enough to convince myself that the main results could be proved along the lines suggested.

**Questions:**

The organization of the introduction should be rethought in order to not have to rely so much on the use of terms that have not yet been properly defined.

**Ethical Concerns:**

["NO or VERY MINOR ethics concerns only"]

**Final Justification:**

I am torn between ratings of 4 and 5 here.

This is a nice result and my main specific issues were presentational which the authors will (hopefully) fix. However, I do not think that I can justify this paper has having a "high impact on at least one sub-area of AI or moderate-to-high impact on more than one area of AI" (realistically, I feel like the impact of this paper is going to be that it will lead to a handful of followup works and be cited a couple dozen times additionally by people who just want to say that this paper had some theoretical results about this issue), as required by the description of the 5 rating. So, despite pushing the limits of the "use sparingly" instruction, I believe I have to keep my official rating at a 4.

**Limitations:**

Yes

**Quality:**

3

**Strengths And Weaknesses:**

Strengths:
* This seems like an interesting new problem direction
* The paper introduces some non-trivial new technical ideas

Weaknesses:
* My biggest issue here is presentational. Many of the main results are introduced *before* the terminology necessary to make sense of them. This is especially bad as they use terms like "noise sensitivity" in non-standard ways. This makes it very hard to follow the introduction without reading it twice. The authors should put more thought into how to organize the introduction so that it is readable.
* I also worry some about the more general applicability of these ideas. The notion of noise sensitivity that they introduce seems to work well for functions that essentially depend on only a few directions, but I worry that for even slightly more complicated functions it will fail badly as the distribution of x and x+z are very far from each other. For example the function 1{|x| > sqrt{d}(1+delta)} will have noise sensitivity almost 1 at almost all Gaussian inputs x, suggesting that this technique has little hope of success to generalize to degree-2 threshold functions.

---

> ### Author Rebuttal · Authors · 2025-07-29
>
> Thank you for suggesting that the introduction be reorganized for clarity. In the final version of this paper, we will state our definitions of isolation probability and noise sensitivity early in the technical overview so that it can be self-contained. We can also use an alternative term such as “perturbation sensitivity” to avoid confusion between our notion and the standard definition of noise sensitivity.
>
> The current work focuses on the highly natural and basic class of halfspaces. Halfspaces are a fundamental class of functions, and there has been a large number of works that exclusively focus on this class of functions. Historically, algorithms designed for halfspaces have frequently yielded algorithms for further hypothesis classes with further technical work, but for this paper we did not directly aim to develop algorithms for other hypothesis classes.
>
> One generalization we had in mind was simple polytopes, which have low boundary volume and low noise sensitivity in the sense presented in this paper. Currently, robustly learning even intersections of two halfspaces remains an open problem, and we think that techniques similar to ours may be useful for this problem.
>
> We agree that the class of low-degree PTFs are a natural target for future work. It is correct that using our definition of local noise sensitivity there is a large volume of points for which the local noise sensitivity is large. However, one could potentially try to sidestep this by considering a noise distribution that uses rescaling similar to the more standard Ornstein–Uhlenbeck process, together with ideas presented in this work.

---

> > ### Comment · Reviewer_mppR · 2025-08-05
> > **Acknowledgement**
> >
> > The willingness to change the presentational issues is appreciated. However, I am not changing my score.

---

### Official Review · Reviewer_d9eS · 2025-07-02

**Clarity:** 4
**Significance:** 3
**Originality:** 3
**Rating:** 4
**Confidence:** 2

**Summary:**

This paper presents an efficient algorithm for adversarially robust learning of halfspaces under subgaussian isotropic log-concave distributions. The algorithm achieves both low classification error and small boundary volume, ensuring robustness to adversarial perturbations. The authors build on polynomial regression techniques, augmenting them with structured rounding and local correction steps to ensure both robustness and efficiency. The results match the best known runtimes for non-robust agnostic learning and provide verifiable robustness guarantees.

**Questions:**

1. The work focuses on subgaussian isotropic log-concave distributions. It would be helpful to understand how critical these assumptions are. Have the authors considered whether the approach or guarantees extend to broader families of distributions? Even a brief discussion on the feasibility or limitations of such relaxations would add value.

2. It would be interesting to see how the proposed techniques perform beyond the theoretical setting, e.g., on synthetic or real data with more complex structure. Experiments validating the theoretical guarantees or exploring tradeoffs in practice would strengthen the work.

3. Another promising direction could be applying the proposed correction and smoothing techniques to existing robust models (e.g., adversarially trained networks). For instance, could the smoothing procedure help improve or measure robustness in practical deep learning models?

**Ethical Concerns:**

["NO or VERY MINOR ethics concerns only"]

**Final Justification:**

In the rebuttal, the authors agreed adding the discussion I recommended for the local and global noise sensitivity.

Regarding my questions, I got a sufficient answer for the first question. For 2 and 3, the authors left further experiments for future work. In my opinion the paper can benefit from some experiments, but it introduce interesting algorithm and theoretical analysis that can stand for its own.

**Limitations:**

yes.

**Quality:**

3

**Strengths And Weaknesses:**

Strengths:

Clarity: The paper is exceptionally well-written and clearly structured. The technical content is explained with clarity, and the motivation behind each algorithmic step is easy to follow. The coverage of related work is thorough and positions the contribution well in the context of existing literature.

Significance and Originality: The combination of noise-sensitivity-aware learning, randomized partitioning, and local correction for achieving robust classification is a compelling and original approach. The contribution is clearly stated and supported by theoretical analysis.

Weaknesses:
One point that could be elaborated is the transition from the standard worst-case adversarial perturbation model (expressed via boundary volume) to average-case robustness notions like noise sensitivity. The intuitive and technical connection between the two robustness notions could benefit from a more explicit discussion, particularly early in the paper (e.g., Section 1.2). Making this bridge more transparent would improve accessibility for readers less familiar with these distinct robustness perspectives.

---

> ### Author Rebuttal · Authors · 2025-07-29
>
> Thank you for suggesting a more thorough discussion of the relationship between adversarial robustness and noise sensitivity. Intuitively, a function can have low global noise sensitivity and not be adversarially robust, when the classification boundary lies near many “isolated” points scattered throughout the domain. Each such isolated region has high local noise sensitivity; thus by using local noise sensitivity as the proxy for local correction, we eliminate these regions. We will add a discussion of this to the final version of this paper.
>
> Regarding Question 1, we repeat the discussion given in response to reviewer 7bYh:
>
> Some distributional assumptions are necessary to evade NP-hardness results, and furthermore some assumptions are necessary for the class of halfspaces itself to be adversarially robust – without anticoncentration, a distribution could place a lot of mass on the classification boundary.
>
> The assumption that the distribution is isotropic is not crucial: if the covariance matrix has all eigenvalues between a pair of positive constants $C_1$ and $C_2$, one can perform a change basis and reduce the problem to the isotropic case (while preserving adversarial robustness). If covariance has eigenvalues very close to zero, halfspaces stop being adversarially robust under such distributions.
>
> Regarding the other assumptions, not much is known about learning halfspaces without these assumptions even in the non-robust improper setting, and not much is known about how learning algorithms perform when the assumptions are slightly violated. If log-concavity is assumed but subgaussianity is not assumed, the best run-time given in [KKMS ‘08] is doubly exponential in $1/\epsilon$ (which is much slower than the algorithms on which our work focuses). No non-trivial algorithms are known if only sub-Gaussianity is assumed.
>
> Regarding Questions 2 and 3, we agree that experimental results would be interesting and consider this an avenue for future work. Analyzing the effects of local correction on other models is an interesting direction for future work as well.

---

### Official Review · Reviewer_3HgY · 2025-07-03

**Clarity:** 3
**Significance:** 2
**Originality:** 2
**Rating:** 4
**Confidence:** 2

**Summary:**

A problem of learning half spaces is considered here. The requirements on the learning algorithm are twofold, first, the learned classifier has larger error than the best halfspace classifier by at most $\epsilon$, and second, the volume of the classifier r-boundary is  $O(r)$. The classifiers r-boundary volume is defined as the probability (over the data distribution) that there is for an input x another point y such that ||x-y|| < r and h(x) and h(y) differ.

First, relaxed learner produces $1/\epsilon^2$-degree polynomial thresholding hypotheses (somewhat uniformly) robust to random (not necessarily adversarial) perturbations via convex programming. Then, a local corrector is used; roughly speaking, it computes probability that a random perturbation changes decision of the classifier. If this probability is high, the label is flipped.

The derived sample complexity then almost matches the sample complexity of the non-robust learning up to polylog factors in the exponent (the sample complexity is exponential in both cases, thus the exponential polylog factor differences are "fine").

**Questions:**

-

**Ethical Concerns:**

["NO or VERY MINOR ethics concerns only"]

**Final Justification:**

-

**Limitations:**

-

**Paper Formatting Concerns:**

-

**Quality:**

3

**Strengths And Weaknesses:**

The paper is very technical but it somewhat successfully tries to be accessible. The considered problem is pretty niche without direct applications in the practice of adversarial robustness (for instance, because of the exponential sample complexity), but it solves the considered problem within a reasonable completeness. The presented techniques seem to be often borrowed from other works, but I am not able to comment if the applications are inovative or straightforward. Some technical novelty is clearly present.

---

> ### Author Rebuttal · Authors · 2025-07-29
>
> Indeed our running time is exponential in $1/\epsilon$, as is all previous work on learning halfspaces up to error $opt+\epsilon$, and thus it is meant for settings where $d$ is the limiting parameter. We note that prior to our work, no algorithm with run-time better than $2^{\Omega(d)}$ was known for the task we consider.
>
> An exponential dependence on $1/\epsilon^2$ in the running time is believed to be necessary due to the statistical query lower bound for agnostically learning halfspaces [Diakonikolas, Kane, Pittas, Zarifis ‘21] which is also exponential in $1/\epsilon^2$. There are also run-time lower bounds based on commonly accepted cryptographic hardness assumptions [Diakonikolas, Kane, Ren ICML 2023].
>
> Regarding the novelty of our techniques:
> - The use of low-degree approximating polynomials for learning is a well-established technique; however this work introduces a proxy for adversarial robustness and shows that polynomial optimization under this constraint is feasible. This type of analysis is new in this work.
> - The use of local correction for learning continues a line of work that is a few years old (introduced in [LRV22]). Previous work uses local correction only to ensure monotonicity of a hypothesis; this is the first application of this technique outside of that setting.
> - The randomized partitioning and rounding strategy is, to the best of our knowledge, new in this work.

---

### Official Review · Reviewer_7bYh · 2025-07-06

**Clarity:** 3
**Significance:** 4
**Originality:** 3
**Rating:** 5
**Confidence:** 4

**Summary:**

This paper presents an efficient algorithm for learning halfspaces that are robust against adversarial perturbations. That is, the probability that a random example is within distance $r$ of its closest point on the other side of the halfspace (called “boundary volume”) scales linearly with $r$. The method is designed for data from subgaussian, isotropic, log-concave distributions. While standard methods like polynomial regression are efficient, they can yield non-robust polynomial threshold functions (PTFs), which also constitutes an improper learner. This new algorithm overcomes the former limitation, producing a classifier with a guaranteed boundary volume of $O(r+\epsilon)$ and a misclassification error of opt+$O(\epsilon)$, where opt denotes the optimal misclassification error. The time and sample complexity of $d^{\tilde{O}(1/\epsilon^2)}$ matches that of the state-of-the-art non-robust algorithms. Technical contributions of the paper include performing $\ell_1$-error polynomial regression under noise sensitivity constraints, a randomized partitioning and rounding method designed to find a classifier that has 3 desired properties simultaneously and finally a local computation algorithm for correction. Specifically, the latter algorithm receives classification queries for a polynomial threshold function and answers according to a robust classifier that is not too far from it. Lastly, the paper introduces the concept of verifiable robustness, showing that under the assumption P=BPP, the output model's robustness can be efficiently and deterministically checked (or with some error probability without making the assumption).
  Minor comment:
Page 6: footnote 3 seems incomplete.

**Questions:**

In theorem 1.1, your guarantees rely on the input distribution being subgaussian, isotropic, and log-concave. How sensitive do you expect the algorithm's performance to be if these assumptions are slightly violated? Have you considered extending these results to more general distributions, perhaps with weaker guarantees?

**Ethical Concerns:**

["NO or VERY MINOR ethics concerns only"]

**Final Justification:**

I am satisfied with the answers to my questions by the reviewers and I am therefore keeping my recommendation for acceptance.

**Limitations:**

yes

**Quality:**

4

**Strengths And Weaknesses:**

The paper introduces novel techniques to obtain a robust version of a fundamental problem in machine learning, without sacrificing sample complexity or computational efficiency. The authors are still using an improper learning algorithm, which is in line with prior work, while a proper learning algorithm would be preferable in general.

---

> ### Author Rebuttal · Authors · 2025-07-29
>
> Thank you for pointing out the incomplete footnote. It should say “The algorithm of [Dan15] also applies only to the Gaussian distribution, and not to general sub-Gaussian log-concave distributions.”
>
> Indeed improperness is a limitation of our algorithm. Agnostic proper learning in this setting is a longstanding open problem.
>
> Some distributional assumptions are necessary to evade NP-hardness results, and furthermore some assumptions are necessary for the class of halfspaces itself to be adversarially robust – without anticoncentration, a distribution could place a lot of mass on the classification boundary.
>
> The assumption that the distribution is isotropic is not crucial: if the covariance matrix has all eigenvalues between a pair of positive constants $C_1$ and $C_2$, one can perform a change basis and reduce the problem to the isotropic case (while preserving adversarial robustness). If covariance has eigenvalues very close to zero, halfspaces stop being adversarially robust under such distributions.
>
> Regarding the other assumptions, not much is known about learning halfspaces without these assumptions even in the non-robust improper setting, and not much is known about how learning algorithms perform when the assumptions are slightly violated. If log-concavity is assumed but subgaussianity is not assumed, the best run-time given in [KKMS ‘08] is doubly exponential in $1/\epsilon$ (which is much slower than the algorithms on which our work focuses). No non-trivial algorithms are known if only subgaussianity is assumed.
>
> The question of slightly relaxing the log-concavity assumption is currently an active area of research, but is not fully understood even for regular (non-robust) agnostic learning. Though there are algorithms that handle distributions $\epsilon$-close in TV distance to specific distributions [Goel, Shetty, Stavropoulos, Vasilyan NeurIPS 2024], there is no non-trivial algorithm known that achieves error $opt+O(\epsilon)$ for all distributions $\epsilon$-close to log-concave in TV distance (outside of the brute-force $2^d$ time algorithms).

---

### Decision · Program_Chairs · 2025-09-17

**Decision:**

Accept (spotlight)

**Comment:**

The authors propose an algorithm for agnostically learning linear threshold classifiers that are adversarially robust. This is a theoretical paper, and it is the first (in a sense efficient) method for solving this problem. The robustness guarantee seems to be weaker  than then standard PAC adversarial learning framework. Nevertheless, the reviewers found the problem setting interesting, the results illuminating, and the proof techniques novel and sophisticated. Overall, this is a strong theory paper that defines and solves a challenging optimization problem.